# Noisy Label Learning with Instance-Dependent Outliers: Identifiability via Crowd Wisdom

**Tri Nguyen**[*]
School of EECS
Oregon State University
Corvallis, Oregon, USA
nguyetr9@oregonstate.edu

**Shahana Ibrahim**[*]
Department of ECE
University of Central Florida
Orlando, Florida, USA
shahana.ibrahim@ucf.edu

**Xiao Fu**
School of EECS
Oregon State University
Corvallis, Oregon, USA
xiao.fu@oregonstate.edu

## Abstract

The generation of label noise is often modeled as a process involving a probability transition matrix (also interpreted as the *annotator confusion matrix*) imposed onto the label distribution. Under this model, learning the "ground-truth classifier"—i.e., the classifier that can be learned if no noise was present—and the confusion matrix boils down to a model identification problem. Prior works along this line demonstrated appealing empirical performance, yet identifiability of the model was mostly established by assuming an instance-invariant confusion matrix. Having an (occasionally) instance-dependent confusion matrix across data samples is apparently more realistic, but inevitably introduces outliers to the model. Our interest lies in confusion matrix-based noisy label learning with such outliers taken into consideration. We begin with pointing out that under the model of interest, using labels produced by only one annotator is fundamentally insufficient to detect the outliers or identify the ground-truth classifier. Then, we prove that by employing a crowdsourcing strategy involving multiple annotators, a carefully designed loss function can establish the desired model identifiability under reasonable conditions. Our development builds upon a link between the noisy label model and a column-corrupted matrix factorization mode—based on which we show that crowdsourced annotations distinguish nominal data and instance-dependent outliers using a low-dimensional subspace. Experiments show that our learning scheme substantially improves outlier detection and the classifier's testing accuracy.

## 1 Introduction

Deep neural networks can easily overfit to noisy labels due to its excessive expressiveness [1,2]. Many strategies have been proposed to avoid negative impacts of label noise when training neural classifiers; see, e.g., noisy label filtering [3–8], robust loss design [9–13], and noise generation modeling (or loss correction) [14–22]. In the last genre, a noisy class label is modeled as a realization of a categorical distribution, which is the ground-truth label distribution being distorted by a *probability transition matrix*. The transition matrix is interpreted as the "confusion matrix" [9, 14, 18] that can effectively

---

[*]Equal contribution.

38th Conference on Neural Information Processing Systems (NeurIPS 2024).

model the annotators' expertise level and the difficulty of annotating each class/sample, and thus is considered intuitive. Under this model, learning the "label noise-free" target neural classifier boils down to identifying the confusion matrix.

The confusion matrix-based models have proven quite useful in practice—algorithms developed in this line of work often exhibits appealing empirical performance; see, e.g., [9,14–17,19–21,23–31]. In addition, these models admit interesting statistical and algebraic structures, leading to plausible results on identifiability of the confusion matrix and/or the "ground-truth classifier"—i.e., the classifier that can be learned if no noisy annotations were present. However, most of the aforementioned early works considered an instance-invariant confusion matrix—i.e., a confusion matrix is not affected by sample features, but only classes—for analytical and computational simplicity. Considering *instance-dependent* confusion models is more realistic, as the sample characteristics, e.g., lightening and resolution of an image, affect the annotation accuracy [32]. The existence of such (at least occasionally occurred) instance-dependent noisy labels inevitably introduces outliers to the instance-invariant confusion models, leading to performance degradation. In general, learning under instance-dependent confusion matrices is heavily ill-posed. Hence, various problem-specific structures were exploited to add regularization terms and constraints; see, e.g., [27,28,33–43]. Nonetheless, unlike the instance-invariant confusion matrix case, identifiability guarantees of the target classifier have been largely under-studied. The lack of theoretical understanding also affects algorithm design—many approaches in this domain had to resort to somewhat ad-hoc treatments with multi-stage training procedures, often involving nontrivial pre- and post-processing; see [27,28,33–35,37,39,40].

**Contributions.** To advance understanding, we consider a model where instance-dependent confusion matrices occur occasionally across the samples, and the rest of data share a common *nominal* confusion matrix. This way, the instance-dependent noisy labels can be regarded as outliers. The model is motivated by the fact that only a proportion of all instances may have a labeling difficulty that significantly deviates from the general population [36,44,45]. Our contributions are as follows:

*(i) Identifiability via Crowdsourcing.* We first show that using the sparsity prior on outliers is insufficient to identify the model of interest, if only one annotator is present. To circumvent this challenge, we propose to employ a *crowd* of annotators—which is the common practice in data labeling [14, 18, 20, 31]. We show that, by incorporating a carefully designed column sparsity constraint to a *coupled cross-entropy* (CCE) loss from crowdsourcing [20, 31] to integrate the annotators' outputs, the outliers can be provably identified. Consequently, the ground-truth classifier can be learned with generalization guarantees.

*(ii) End-to-end One-Stage Implementation.* Our approach features a one-stage continuous optimization-based implementation. To be specific, our proposed learning loss allows to approximate the column-sparsity constraint using a smoothed nonconvex $\ell_p$ (where $0 < p \le 1$) function based regularization (see [46]). This way, the overall loss is differentiable and can be readily tackled by off-the-shelf optimizers. This is in contrast to many existing methods that involve multiple stages (e.g., [27, 28, 33–35, 37, 39, 40])—and is arguably easier to implement.

We evaluate the proposed method over a number real datasets that are annotated by machine and human annotators under various conditions and observed nontrivial improvements of testing accuracy.

**Notation.** The notations are summarized in the supplementary material in Sec. A.

## 2   Problem Statement

**The Confusion Model and Learning Goal.** Consider $N$ data items $\{\boldsymbol{x}_n\}_{n=1}^{N}$ from $K$ classes. Here, $\boldsymbol{x}_n \in \mathbb{R}^D$ represents the feature vector of the $n$th data item. Let $\{y_n\}_{n=1}^{N}$ be the set of ground-truth labels, where $y_n \in [K]$. Assume that $y_n$'s are unobserved. Instead, we observe the "noisy" version $\{\widehat{y}_n\}_{n=1}^{N}$. The label $\widehat{y}_n \in [K]$ is noisy due to various reasons, e.g., the lack of expertise of the annotator. In this setting, our goal is to learn a performance-guaranteed classifier using the data items $\{\boldsymbol{x}_n\}_{n=1}^{N}$, and noisy labels $\{\widehat{y}_n\}_{n=1}^{N}$. We consider the following expression of $\Pr(\widehat{y}_n = k|\boldsymbol{x}_n)$:

$$\Pr(\widehat{y}_n = k|\boldsymbol{x}_n) = \sum_{k'=1}^{K} \Pr(\widehat{y}_n = k|y_n = k', \boldsymbol{x}_n)\Pr(y_n = k'|\boldsymbol{x}_n). \tag{1}$$

Note that $\Pr(y_n = k|\boldsymbol{x}_n)$ is the *ground-truth label distribution* given the sample $\boldsymbol{x}_n$. This is also the distribution that the *target classifier* wishes to learn from. We represent the *ground-truth classifier*

using a function $\boldsymbol{f}^{\natural} : \mathbb{R}^D \to \mathbb{R}^K$ such that $[\boldsymbol{f}^{\natural}(\boldsymbol{x}_n)]_k \triangleq \Pr(y_n = k | \boldsymbol{x}_n)$. We call $\boldsymbol{f}^{\natural}$ the ground-truth classifier as it is the function that we aim to learn—and it can be learned under ideal conditions (e.g., when $N = \infty$ and the learner is a universal approximator), given that no noise is present. We define $\boldsymbol{T}^{\natural} : \mathbb{R}^D \to \mathbb{R}^{K \times K}$ such that $[\boldsymbol{T}^{\natural}(\boldsymbol{x}_n)]_{k,k'} \triangleq \Pr(\widehat{y}_n = k | y_n = k', \boldsymbol{x}_n), \forall \boldsymbol{x}_n$. That is, $\boldsymbol{T}^{\natural}(\boldsymbol{x}_n)$ is the *label transition matrix* or *confusion matrix* of sample $\boldsymbol{x}_n$. Let $[\boldsymbol{g}_n^{\natural}]_k \triangleq \Pr(\widehat{y}_n = k | \boldsymbol{x}_n)$. Accordingly, the noisy label generation process for $\boldsymbol{x}_n \overset{\text{i.i.d.}}{\sim} \mathcal{D}_{\boldsymbol{x}}$ is modeled as follows:

$$\boldsymbol{g}_n^{\natural} = \boldsymbol{T}^{\natural}(\boldsymbol{x}_n) \boldsymbol{f}^{\natural}(\boldsymbol{x}_n), \tag{2a}$$

$$\widehat{y}_n \sim \text{categorical}(\boldsymbol{g}_n^{\natural}), \tag{2b}$$

where categorical$(\boldsymbol{g})$ denotes the $K$-dimensional categorical distribution. Per the physical meaning of $\boldsymbol{T}^{\natural}(\boldsymbol{x}_n)$ and $\boldsymbol{f}^{\natural}(\boldsymbol{x}_n)$, we have $\mathbf{1}^\top \boldsymbol{T}^{\natural}(\boldsymbol{x}_n) = \mathbf{1}^\top, \boldsymbol{T}^{\natural}(\boldsymbol{x}_n) \geq \mathbf{0}$ and $\mathbf{1}^\top \boldsymbol{f}^{\natural}(\boldsymbol{x}_n) = 1, \boldsymbol{f}^{\natural}(\boldsymbol{x}_n) \geq \mathbf{0}$, for all $n$. Under this model, the main goal is to learn $\boldsymbol{f}^{\natural}$ from $\{\boldsymbol{x}_n\}_{n=1}^N$ and $\{\widehat{y}_n\}_{n=1}^N$.

**Identifiability under Instance-Invariant $\boldsymbol{T}^{\natural}(\boldsymbol{x})$.** From (2a), it becomes apparent that learning $\boldsymbol{f}^{\natural}$ is not a straightforward task. Even if $\boldsymbol{g}_n^{\natural}$ is observed (which is not), it is hard to identify $\boldsymbol{f}^{\natural}(\boldsymbol{x}_n)$ or $\boldsymbol{T}^{\natural}(\boldsymbol{x}_n)$ from the product $\boldsymbol{g}_n^{\natural} = \boldsymbol{T}^{\natural}(\boldsymbol{x}_n) \boldsymbol{f}^{\natural}(\boldsymbol{x}_n)$. To tackle the identifiability challenge, a popular approach is to simplify (2) by assuming that all instances have the same confusion matrix, i.e., $\boldsymbol{T}^{\natural}(\boldsymbol{x}_n) = \boldsymbol{A}^{\natural}$ with a certain $\boldsymbol{A}^{\natural} \in \mathbb{R}^{K \times K}$ for all $n$ [16, 18, 19, 21, 47, 48]. Under this assumption, many used the "loss correction" based formulation, e.g., [9, 16, 19, 21, 48]. The loss correction idea modifies the training loss of the classifier by taking the confusion matrix into consideration, e.g.,

$$\underset{\boldsymbol{A} \in \mathcal{A}, \boldsymbol{f} \in \mathcal{F}}{\text{minimize}} -\frac{1}{N} \sum_{n=1}^N \sum_{k=1}^K [\widehat{\boldsymbol{y}}_n]_k \log[\boldsymbol{A}\boldsymbol{f}(\boldsymbol{x}_n)]_k, \tag{3}$$

where the objective is a modified *cross-entropy* (CE) loss. The "noise correction" term $\boldsymbol{A}$ and $\boldsymbol{f}$ are used to learn $\boldsymbol{A}^{\natural}$ and $\boldsymbol{f}^{\natural}$, respectively, $\widehat{\boldsymbol{y}}_n \in \{0, 1\}^K$ denotes the one-hot encoding of the noisy label $\widehat{y}_n$, and $\mathcal{A}$ and $\mathcal{F}$ denote appropriate constraint sets—i.e., $\mathcal{A} = \{\boldsymbol{A} \in \mathbb{R}^{K \times K} \mid \mathbf{1}^\top \boldsymbol{A} = \mathbf{1}^\top, \boldsymbol{A} \geq \mathbf{0}\}$ and $\mathcal{F}$ is a certain deep neural network function class whose outputs reside in the probability simplex. Note that the objective is often used together with regularization and additional constraints of $\boldsymbol{A}$ for various purposes; see, e.g., [19, 21, 30]. When $N \to \infty$, the CE term enforces the learned $\boldsymbol{A}$ and $\boldsymbol{f}$ to satisfy $\boldsymbol{g}_n^{\natural} = \boldsymbol{A}\boldsymbol{f}(\boldsymbol{x}_n)$ for all $n$ [19]. Note that $\widetilde{\boldsymbol{G}} = [\boldsymbol{g}_1^{\natural}, \dots, \boldsymbol{g}_N^{\natural}]$ can be expressed as $\widetilde{\boldsymbol{G}} = \boldsymbol{A}^{\natural} [\boldsymbol{f}^{\natural}(\boldsymbol{x}_1), \dots, \boldsymbol{f}^{\natural}(\boldsymbol{x}_N)] = \boldsymbol{A}^{\natural} \boldsymbol{F}^{\natural}$, where $\boldsymbol{F}^{\natural} = [\boldsymbol{f}^{\natural}(\boldsymbol{x}_1), \dots, \boldsymbol{f}^{\natural}(\boldsymbol{x}_N)]$. Consequently, Eq. (3) can be understood as learning $\boldsymbol{A}$ and $\boldsymbol{F}$ such that $\widetilde{\boldsymbol{G}} \approx \boldsymbol{A}\boldsymbol{F}$. As both $\boldsymbol{A}^{\natural}$ and $\boldsymbol{F}^{\natural}$ are nonnegative matrices (due to their physical meaning), the identifiability of $\boldsymbol{F}^{\natural}$ can be connected to uniqueness of the *nonnegative matrix factorization* (NMF) model $\widetilde{\boldsymbol{G}} = \boldsymbol{A}^{\natural} \boldsymbol{F}^{\natural}$; see [19, 21, 45].

These are interesting developments, yet the key assumption $\boldsymbol{T}^{\natural}(\boldsymbol{x}_n) = \boldsymbol{A}^{\natural}$ for all $n$ appears to be overly stringent. As mentioned, it makes sense to believe that at least a proportion of samples have instance-dependent $\boldsymbol{T}^{\natural}(\boldsymbol{x}_n)$'s [36, 44, 45, 49]. Not considering such samples may cause performance degradation.

**Instance-Dependent Confusion-Induced Outliers.** When the instance-dependent confusion happens sparingly instead of overwhelmingly, we can re-express $\boldsymbol{T}^{\natural}(\boldsymbol{x}_n)$ using the following decomposition:

$$\boldsymbol{T}^{\natural}(\boldsymbol{x}_n) = \boldsymbol{A}^{\natural} + \boldsymbol{E}^{\natural}(\boldsymbol{x}_n), \tag{4}$$

where $\boldsymbol{A}^{\natural} \in \mathbb{R}^{K \times K}$ represents an instance-independent (class-dependent) confusion matrix—which is the nominal confusion matrix that reflects the general annotation difficulty of the dataset. The term $\boldsymbol{E}^{\natural}(\boldsymbol{x}_n)$ represents the instance-dependent "perturbation". For many $n$'s, $\boldsymbol{E}^{\natural}(\boldsymbol{x}_n) = \mathbf{0}$. When $\boldsymbol{E}^{\natural}(\boldsymbol{x}_n) \neq \mathbf{0}$, we have $(\boldsymbol{A}^{\natural} + \boldsymbol{E}^{\natural}(\boldsymbol{x}_n))\boldsymbol{f}^{\natural}(\boldsymbol{x}_n) = \boldsymbol{A}^{\natural}\boldsymbol{f}^{\natural}(\boldsymbol{x}_n) + \boldsymbol{e}_n^{\natural}$, where $\boldsymbol{e}_n^{\natural} = \boldsymbol{E}^{\natural}(\boldsymbol{x}_n)\boldsymbol{f}^{\natural}(\boldsymbol{x}_n)$. Using (4), the model in (2a) can be expressed as follows:

$$\boldsymbol{g}_n^{\natural} = \begin{cases} \boldsymbol{A}^{\natural}\boldsymbol{f}^{\natural}(\boldsymbol{x}_n) + \boldsymbol{e}_n^{\natural}, & \forall n \in \mathcal{I} \\ \boldsymbol{A}^{\natural}\boldsymbol{f}^{\natural}(\boldsymbol{x}_n), & \forall n \in \mathcal{I}^c, \end{cases} \tag{5}$$

where $\mathcal{I}$ and $\mathcal{I}^c$ represent the instance index set where $\boldsymbol{E}^{\natural}(\boldsymbol{x}_n) \neq \mathbf{0}$ and its complement, respectively. In other words, $\mathcal{I}$ is the outlier index set.

A remark is that from this point on we will ignore the structure $\boldsymbol{e}_n^{\natural} = \boldsymbol{E}^{\natural}(\boldsymbol{x}_n)\boldsymbol{f}^{\natural}(\boldsymbol{x}_n)$ of the outliers. Disregarding the structure comes with some losses. For example, this way, our method is not able

to learn $\boldsymbol{E}^{\natural}(\boldsymbol{x}_n)$ that could be of interest. Nonetheless, considering $\boldsymbol{E}^{\natural}(\boldsymbol{x}_n)$ renders extra modeling and computational burdens. Our treatment simplifies the subsequent analytical and computational developments, particularly, algorithm design. In addition, other types of outliers and anomalies can also be handled by the proposed approach due to the unstructured treatment.

**Challenge of Outlier Detection.** Under the model in (5), the natural idea is to first identify $\mathcal{I}$ and remove the impacts of outliers. Our first attempt is to explicitly model $\boldsymbol{e}_n^{\natural}$ and modify (3) as follows:

$$\underset{\boldsymbol{A}\in\mathcal{A},\{\boldsymbol{e}_n\in\mathcal{E}\},\boldsymbol{f}\in\mathcal{F}}{\text{minimize}} \quad -\frac{1}{N}\sum_{n=1}^{N}\sum_{k=1}^{K}[\widehat{\boldsymbol{y}}_n]_k \log\left[\boldsymbol{A}\boldsymbol{f}(\boldsymbol{x}_n)+\boldsymbol{e}_n\right]_k,$$

$$\text{subject to } \sum_{n=1}^{N}\mathbb{1}\left\{\|\boldsymbol{e}_n\|_2 > 0\right\} \le C,$$

where $\mathcal{E} = \{\boldsymbol{e}\in\mathbb{R}^K \mid \boldsymbol{1}^{\top}\boldsymbol{e} = 0\}$ is the constraint set for $\boldsymbol{e}_n$'s (due to the probability simplex constraints on the model parameters in (5)) and $C$ is a scalar that is an estimation of $|\mathcal{I}|$. The idea is to use the prior knowledge that $\boldsymbol{e}_n$ does not occur overwhelmingly as constraint. The hope is that the solution of (6) can detect $\mathcal{I}$, thereby enabling accurate estimation of $\boldsymbol{A}^{\natural}$ and $\boldsymbol{f}^{\natural}$. However, the following fact reveals a conflicting insight:

**Fact 2.1.** *Assume that all $\boldsymbol{f}\in\mathcal{F}$ are universal function approximators, that $\mathrm{rank}(\boldsymbol{A}^{\natural}) = K$, and that $N\to\infty$. Suppose that $\mathcal{I}\neq\emptyset$. Optimal solutions of Problem* (6) *can attain the minimal value of the objective function and satisfy the sparsity constraint without detecting any outliers. One such trivial solution $(\boldsymbol{A}^{\star},\boldsymbol{f}^{\star},e_n^{\star})$ is $\boldsymbol{A}^{\star} = \boldsymbol{I}$, $\boldsymbol{f}^{\star}(\boldsymbol{x}) = \boldsymbol{A}^{\natural}\boldsymbol{f}^{\natural}(\boldsymbol{x}_n)+\boldsymbol{e}_n^{\natural}$, and $\boldsymbol{e}_n^{\star} = \boldsymbol{0}$ for all $n$.*

This fact is easy to show (see Appendix F), yet it highlights a somewhat unexpected issue in confusion matrix-based noisy label learning: If only one nominal confusion matrix $\boldsymbol{A}^{\natural}$ is present (i.e., only one annotator is employed), it does not suffice to recover the ground-truth $\boldsymbol{f}^{\natural}$ when there exists instance-dependent outliers (or any other types of outliers) under the model in (5)—under the condition that $\boldsymbol{f}$ is a universal function approximator. The practical implication is undesirable: as deep neural networks are powerful function approximators and are usually very expressive, Fact 2.1 means that the learned neural networks easily overfit to outliers, even if the sparsity prior on outliers is explicitly used in the loss function.

## 3  Proposed Approach

**Intuition: Exploiting Crowd-induced Subspace.** To get around the issues illustrated in Fact 2.1, our idea is to "create" a low-dimensional subspace where the nominal data reside while the outliers are likely far away from. To this end, we propose a *crowdsourcing* approach. Consider the scenario where $M$ annotators label the dataset $\{\boldsymbol{x}_n\}_{n=1}^{N}$. The noisy label provided by an annotator $m$ for the data item $\boldsymbol{x}_n$ is denoted as $\widehat{y}_n^{(m)}\in[K]$ and follows the generation model as follows:

$$\widehat{y}_n^{(m)} \sim \mathsf{categorical}(\boldsymbol{g}_n^{\natural(m)}), \quad \boldsymbol{g}_n^{\natural(m)} = \boldsymbol{A}_m^{\natural}\boldsymbol{f}^{\natural}(\boldsymbol{x}_n)+\boldsymbol{e}_n^{\natural(m)}, \forall m, n, \tag{6}$$

where $\boldsymbol{A}_m^{\natural}$ is the annotator $m$'s confusion matrix and $\boldsymbol{e}_n^{\natural(m)}$ is the outlier term induced by annotator $m$'s instance-dependent confusion.

Putting together, we have the following expression:

$$\boldsymbol{G}^{\natural} = \begin{bmatrix} \boldsymbol{g}_1^{\natural(1)} & \cdots & \boldsymbol{g}_N^{\natural(1)} \\ \vdots & \ddots & \vdots \\ \boldsymbol{g}_1^{\natural(M)} & \cdots & \boldsymbol{g}_N^{\natural(M)} \end{bmatrix} = \begin{bmatrix} \boldsymbol{A}_1^{\natural}\boldsymbol{f}^{\natural}(\boldsymbol{x}_1)+\boldsymbol{e}_1^{\natural(1)} & \cdots & \boldsymbol{A}_1^{\natural}\boldsymbol{f}^{\natural}(\boldsymbol{x}_N)+\boldsymbol{e}_N^{\natural(1)} \\ \vdots & \ddots & \vdots \\ \boldsymbol{A}_M^{\natural}\boldsymbol{f}^{\natural}(\boldsymbol{x}_1)+\boldsymbol{e}_1^{\natural(M)} & \cdots & \boldsymbol{A}_M^{\natural}\boldsymbol{f}^{\natural}(\boldsymbol{x}_N)+\boldsymbol{e}_N^{\natural(M)} \end{bmatrix}$$
$$\iff \boldsymbol{G}^{\natural} = \boldsymbol{W}^{\natural}\boldsymbol{F}^{\natural} + \boldsymbol{E}^{\natural}, \tag{7}$$

where $\boldsymbol{F}^{\natural}$ is defined as before, and

$$\boldsymbol{W}^{\natural} = [(\boldsymbol{A}_1^{\natural})^{\top},\ldots,(\boldsymbol{A}_M^{\natural})^{\top}]^{\top}, \quad \boldsymbol{e}_n^{\natural} = [(\boldsymbol{e}_n^{\natural(1)})^{\top},\ldots,(\boldsymbol{e}_n^{\natural(M)})^{\top}]^{\top}. \tag{8}$$

Denote $\mathcal{I}^{(m)}$ as the index set where $\boldsymbol{e}_n^{\natural(m)}\neq\boldsymbol{0}$. Then $\mathcal{I} = \mathcal{I}^{(1)}\cup\ldots\cup\mathcal{I}^{(M)}$ stands for the nonzero column support of $\boldsymbol{E}^{\natural}$.

From (7), one can see that now the "data columns", i.e., the columns of the matrix $\boldsymbol{G}^\natural$, live in a high-dimensional space, i.e., $\mathbb{R}^{MK}$. However, the nominal data part $\boldsymbol{W}^\natural \boldsymbol{F}^\natural$ resides in a $K$-dimensional subspace $\mathrm{range}(\boldsymbol{W}^\natural)$ where $K \ll MK$, if multiple annotators are employed. More importantly, $\boldsymbol{e}_j^\flat$ for $j \in \mathcal{I}$ is an $MK$-dimensional vector and is unlikely to be inside $\mathrm{range}(\boldsymbol{W}^\natural)$. With this geometry, it is much more likely that one can separate the outliers from the nominal data.

Next, we will use the above intuition to build a learning loss. We will take into consideration of practical aspects, e.g., missing annotations and finite dataset size $N$.

**Proposed Identification Criterion & Analysis.** In this section, we connect our proposed idea to a practically convenient, end-to-end identification criterion and provide identifiability guarantees for the desired model parameters. We consider the following empirical risk minimization under the crowdsourcing model in (6):

$$\underset{\{\boldsymbol{A}_m \in \mathcal{A}\}, \{\boldsymbol{e}_n^{(m)} \in \mathcal{E}\}, \boldsymbol{f} \in \mathcal{F}}{\text{minimize}} \quad \mathsf{L}_{\mathsf{ce}} \triangleq -\frac{1}{S} \sum_{(m,n) \in \mathcal{S}} \sum_{k=1}^{K} [\widehat{\boldsymbol{y}}_n^{(m)}]_k \log \left[ \boldsymbol{A}_m \boldsymbol{f}(\boldsymbol{x}_n) + \boldsymbol{e}_n^{(m)} \right]_k, \tag{9a}$$

$$\text{subject to} \quad \sum_{n=1}^{N} \mathbb{1} \left\{ \sum_{m=1}^{M} \|\boldsymbol{e}_n^{(m)}\|_2 > 0 \right\} \leq C, \tag{9b}$$

where $\mathcal{S} \subseteq [M] \times [N]$ denotes the set of observed noisy labels indexed by $(m, n)$ with $S = |\mathcal{S}|$ (note that all data items may not be labeled by an annotator), $\widehat{\boldsymbol{y}}_n^{(m)}$ is the one-hot encoding of the annotator-provided label $\widehat{y}_n^{(m)}$, and $C$ is an estimate of $|\mathcal{I}|$. Here, the constraint sets $\mathcal{F}$, $\mathcal{A}$, and $\mathcal{E}$ are as defined before. If $\boldsymbol{e}_n^{(m)}$ is not considered, the objective function (9a) is sometimes referred to as *coupled cross-entropy minimization* (CCEM) in the end-to-end crowdsourcing literature [20, 30, 31, 50]. CCEM has not been used together with the outlier detection constraint (9b)—and the theoretical characterizations of the constrained formulation is unknown.

We will use the following notations in our analysis. We use $\mathcal{I} \subseteq [N]$ to denote index set of the outliers, i.e., $\mathcal{I} = \{n \mid \boldsymbol{e}_n^\flat \neq \boldsymbol{0}\}$. We also consider that the function class in our learning problem, i.e., $\mathcal{F}$, has a complexity measure $\mathscr{R}_\mathcal{F}$. In particular, we adopt the so-called spectral-complexity upper bound of the function class $\mathcal{F}$ [51]; see Lemma C.4 in the Appendix.

We first establish that the criterion (9) recovers the ground-truth $\boldsymbol{g}_n^{\natural(m)}$'s with a reasonable accuracy. Specifically, we hope to characterize the average estimation accuracy of $\widehat{\boldsymbol{g}}_n^{(m)}$'s where $\widehat{\boldsymbol{g}}_n^{(m)} = \widehat{\boldsymbol{A}}_m \widehat{\boldsymbol{f}}(\boldsymbol{x}_n) + \widehat{\boldsymbol{e}}_n^{(m)}$ and $\widehat{\boldsymbol{A}}_m$'s, $\widehat{\boldsymbol{e}}_n^{(m)}$'s and $\widehat{\boldsymbol{f}}$ are estimated via solving (9):

**Lemma 3.1.** *Assume that the observed index pairs $(m, n) \in \mathcal{S}$ are sampled uniformly at random with replacement. Also assume $\boldsymbol{f}^\natural \in \mathcal{F}$, $|\mathcal{I}| \leq C \leq N/2$, and that each $[\boldsymbol{A}_m \boldsymbol{f}(\boldsymbol{x}_n) + \boldsymbol{e}_n^{(m)}]_k$, $\forall \boldsymbol{A}_m \in \mathcal{A}, \forall \boldsymbol{e}_n^{(m)} \in \mathcal{E}, \forall \boldsymbol{f} \in \mathcal{F}$ are lower bounded by $(1/\beta)$ for a certain $\beta > 0$. Then, with probability greater than $1 - \delta$, the following holds:*

$$\frac{1}{NM} \sum_{n=1}^{N} \sum_{m=1}^{M} \|\boldsymbol{g}_n^{\natural(m)} - \widehat{\boldsymbol{g}}_n^{(m)}\|_2^2 \leq \epsilon_{\boldsymbol{g}}(S), \quad \epsilon_{\boldsymbol{g}}(S) = \mathcal{O}\left( \beta \mathfrak{R}_\mathcal{S} \log S / \sqrt{S} + \log(\beta) \sqrt{\log(1/\delta)} / \sqrt{S} \right),$$

*where $\mathfrak{R}_\mathcal{S}^2 = K^2 \log(K) + K\|\boldsymbol{X}\|_\mathrm{F}^2 \mathscr{R}_\mathcal{F} / S + CMK \log \sqrt{NM}$ and $\boldsymbol{X} = [\boldsymbol{x}_{n_1}, \dots, \boldsymbol{x}_{n_S}]$ are the annotated samples, in which $(m_s, n_s) \in \mathcal{S}$ for $s = 1, \dots, S$.*

The proof is in Appendix C, which uses a similar idea as [31, Proposition 1] but takes into consideration of the outliers. Lemma 3.1 reveals that the criterion (9) essentially recovers the complete matrix $\boldsymbol{G}^\natural$ [cf. Eq, (7)] from the noisy and incomplete observations of its entries, when $S/C$ is sufficiently large. To proceed, we will need a suite of assumptions and definitions:

**Assumption 3.2.** The outliers satisfy $\boldsymbol{e}_i^\flat \notin \mathrm{range}(\boldsymbol{W}^\natural), \forall i \in \mathcal{I}$, where $\mathrm{rank}(\boldsymbol{W}^\natural) = K$.

Assumption 3.2 presents the key condition to disassociate instance-dependent and instance-independent confusions. As discussed after Eq. (8), having a larger $M$ would decrease the possibility that any $\boldsymbol{e}_n^\flat$ belongs to $\mathrm{range}(\boldsymbol{W}^\natural)$. We also characterize the "quantity" that $\boldsymbol{e}_n^\flat$ is perturbed from $\mathrm{range}(\boldsymbol{W}^\natural)$ in the following definition:

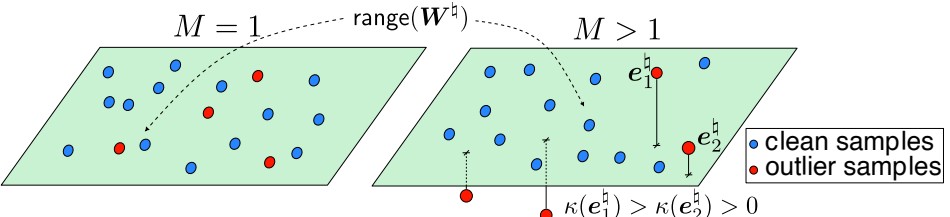

Figure 1: An illustration of outliers and nominal data items w.r.t. $\mathrm{range}(\boldsymbol{W}^\natural)$: (left) when $M = 1$, $\boldsymbol{e}^\natural \in \mathrm{range}(\boldsymbol{W}^\natural)$, (right) while $M > 1$, it is highly likely that $\boldsymbol{e}^\natural \neq \mathrm{range}(\boldsymbol{W}^\natural)$. In addition, the measure $\kappa(\boldsymbol{e})$ is larger for outliers that are farther from $\mathrm{range}(\boldsymbol{W}^\natural)$.

**Definition 3.3.** The *outlier impact score* $\kappa(\boldsymbol{e}_i^\natural)$ of data $i \in \mathcal{I}$ is defined as

$$\kappa(\boldsymbol{e}_i^\natural) \triangleq \min_{\substack{\boldsymbol{W} \in \mathcal{W}; \boldsymbol{h}, \boldsymbol{h}_i \in \mathcal{E}, \\ \mathcal{L} \subset \mathcal{I}^c, |\mathcal{L}| = N - 2|\mathcal{I}|}} \left( \|\boldsymbol{W}^\natural \boldsymbol{F}^\natural(:, \mathcal{L}) - \boldsymbol{W}\boldsymbol{H}\|_{\mathrm{F}}^2 + \|\boldsymbol{W}^\natural \boldsymbol{f}_i^\natural + \boldsymbol{e}_i^\natural - \boldsymbol{W}\boldsymbol{h}\|^2 \right) \qquad (10)$$

where $\mathcal{W} = \left\{ \boldsymbol{W} = [\boldsymbol{A}_1^\top, \ldots, \boldsymbol{A}_M^\top]^\top \mid \boldsymbol{A}_m \in \mathcal{A} \right\}$.

A larger score $\kappa(\boldsymbol{e}_i^\natural)$ implies it is easier for our criterion to distinguish the outliers from the nominal data. Fig. 1 illustrates the geometry that we rely on to detect outliers. One can notice that when $M = 1$, we always have $\kappa(\boldsymbol{e}_i^\natural) = 0$, as the outlier satisfies $\boldsymbol{e}_i^\natural \in \mathrm{range}(\boldsymbol{W}^\natural)$.

In addition, we consider the following assumptions to assist identifying $\boldsymbol{A}^\natural$ and $\boldsymbol{f}^\natural$:

**Assumption 3.4** (**Class Specialists and Anchor Points**). Assume that the following conditions hold:

    a. There exists a near-class specialist for each class $k$, i.e., $\forall k \in [K], \exists m_k \in [M]$ such that $\|\boldsymbol{A}_{m_k}^\natural(k, :) - \overline{\boldsymbol{e}}_k^\top\| \leq \xi_1$, where $\overline{\boldsymbol{e}}_k$ is a unit vector.

    b. There exists a near-anchor point sample for each class $k$ in the dataset, i.e., $\forall k \in [K], \exists n_k \in \mathcal{I}^c$ such that $\|\boldsymbol{f}^\natural(\boldsymbol{x}_{n_k}) - \overline{\boldsymbol{e}}_k^\top\| \leq \xi_2$, where $\overline{\boldsymbol{e}}_k$ is a unit vector.

Assumption 3.4.a. is sometimes used in the crowdsourcing literature (e.g., [52]) to characterize the expertise of annotators. Assumption 3.4.b. is often seen in loss correction based noisy label learning; see, e.g., [16, 21, 27]. Under Assumptions 3.2 and 3.4, we have the following result:

**Theorem 3.5** (**Identifiability and Generalization**). *Let* $(\{\widehat{\boldsymbol{A}}_m\}, \{\widehat{\boldsymbol{e}}_n^{(m)}\}, \widehat{\boldsymbol{f}})$ *be any optimal solution of* (9) *with* $\widehat{\mathcal{I}} = \left\{ n \in [N] \mid \widehat{\boldsymbol{e}}_n = [(\widehat{\boldsymbol{e}}_n^{(1)})^\top, \ldots, (\widehat{\boldsymbol{e}}_n^{(M)})^\top]^\top \neq \boldsymbol{0} \right\}$. *Suppose that the conditions in Lemma 3.1 holds, that we set* $C = |\mathcal{I}|$, *and that Assumption 3.4 holds with* $\xi_1, \xi_2 \leq 1/K$. *Denote* $\sigma$ *as the upper bound* $\sigma_{\max}(\boldsymbol{A}_m^\natural) \leq \sigma$, $\forall m$. *In addition, assume that* $\forall \mathcal{L} \subset \mathcal{I}^c, |\mathcal{L}| = N - 2|\mathcal{I}^c|$, $\mathrm{rank}(\boldsymbol{F}^\natural(:, \mathcal{L})) = K$. *Then, for some* $\alpha > 0$, *and* $S > S_0$ *where* $S_0$ *as the smallest integer such that* $\kappa(\boldsymbol{e}_i^\natural) \geq \epsilon_{\boldsymbol{g}}(S_0)$, $\forall i \in \mathcal{I}$, *the following result holds with probability greater than* $1 - 2/S - K/T^\alpha$:

$$\mathbb{E}_{\boldsymbol{x} \sim \mathcal{D}_{\boldsymbol{x}}} \left[ \min_{\boldsymbol{\Pi}} \|\widehat{\boldsymbol{f}}(\boldsymbol{x}) - \boldsymbol{\Pi}^\top \boldsymbol{f}^\natural(\boldsymbol{x})\|_2^2 \right] \leq K(\eta + \xi_1 + \xi_2),$$

$$\min_{\boldsymbol{\Pi}} \|\widehat{\boldsymbol{A}}_m - \boldsymbol{A}_m^\natural \boldsymbol{\Pi}\|_{\mathrm{F}}^2 = K\sigma^2(\eta + \xi_1 + \xi_2), \ \forall m,$$

*where* $\eta^2 = \mathcal{O}\left( \beta M T^\alpha / \sqrt{S} \left( \sqrt{M \log S} + (\|\boldsymbol{X}\|_{\mathscr{R}_\mathcal{F}})^{0.25} \right) \right)$, $\boldsymbol{\Pi}$ *a permutation matrix,* $\boldsymbol{X}$ *is defined as before, and* $T = N - |\mathcal{I}|$. *In addition, we have exact outlier detection, i.e.,* $\widehat{\mathcal{I}} = \mathcal{I}$.

The proof is in Appendix D. We should mention that we set $C = |\mathcal{I}|$ for notation simplicity. With a notation-wise slightly more complicated definition of $\mathcal{L}$, the same proof holds under $C \geq |\mathcal{I}|$, which leads to $T = N - C$ and $\mathcal{I} \subseteq \widehat{\mathcal{I}}$. That is, over-estimated $C$ still enables identifying $\mathcal{I}$ with the price of discarding some nominal samples. While the criterion (9) offers the desired identifiability, it requires the presence of class-specialist annotators and the anchor data points (c.f. Assumption 3.4), which are considered relatively restrictive [19, 31]. In [19], the CE loss was combined with a simplex volume minimization-based regularization to establish identifiability of $\boldsymbol{f}^\natural$ under more relaxed conditions, namely, the *sufficiently scattered condition* (SSC) from the NMF literature [53]. We show that this is also viable in the presence of outliers:

**Theorem 3.6 (Enhanced Identifiability).** *Let $(\{\widehat{\boldsymbol{A}}_m\}, \{\widehat{\boldsymbol{e}}_n^{(m)}\}, \widehat{\boldsymbol{f}})$ be any optimal solution of* (9) *with $\widehat{\boldsymbol{W}} \triangleq [\widehat{\boldsymbol{A}}_1^\top, \ldots, \widehat{\boldsymbol{A}}_M^\top]^\top$ admitting the minimum volume of $\mathrm{conv}\{\widehat{\boldsymbol{W}}\}$ (i.e., the simplex spanned by its columns) over all possible optimal solutions. Assume that we set $C = |\mathcal{I}|$ and that $\boldsymbol{F}^\natural(:, \mathcal{I}^c)$ satisfies the SSC. Under Assumptions 3.2 and the same conditions used in Lemma 3.1, the following result holds with probability greater than $1 - \delta$, when $S$ grows to infinity:*

$$\underset{\boldsymbol{x} \sim \mathcal{D}_{\boldsymbol{x}}}{\mathbb{E}} \left[ \|\widehat{\boldsymbol{f}}(\boldsymbol{x}) - \boldsymbol{\Pi}^\top \boldsymbol{f}^\natural(\boldsymbol{x})\|_2^2 \right] = \mathcal{O}(|\mathcal{I}^c|^{-5/8} \, (2\|\boldsymbol{X}(:, \mathcal{I}^c)\|_{\mathrm{F}} \mathscr{R}_{\mathcal{F}})^{\frac{1}{4}} + \sqrt{\log(1/\delta)/|\mathcal{I}^c|}).$$

*In addition, we have $\|\widehat{\boldsymbol{A}}_m - \boldsymbol{A}_m^\natural \boldsymbol{\Pi}\|_{\mathrm{F}} = 0, \forall m$ and $\widehat{\mathcal{I}} = \mathcal{I}$.*

The proof is relegated to Appendix E, which also holds for $C \geq |\mathcal{I}|$ with slight modifications as discussed before. Theorem 3.6 shows that the target classifier can be accurately estimated in the presence of instance-dependent outliers, without needing anchor samples or class specialists.

**Implementation.** Our learning losses allows relatively easy implementation and optimization. We consider the following regularized criterion:

$$\underset{\boldsymbol{A}_m \in \mathcal{A}, \boldsymbol{e}_n^{(m)} \in \mathcal{E}, \boldsymbol{f} \in \mathcal{F}}{\text{minimize}} \quad \mathsf{L}_{\mathsf{ce}} + \mu_1 \mathsf{L}_{\mathsf{outlier}} + \mu_2 \mathsf{L}_{\mathsf{vol}}.$$

Note that we use $\mathsf{L}_{\mathsf{outlier}} = \sum_{n=1}^N (\sum_{m=1}^M \|\boldsymbol{e}_n^{(m)}\|_2^2 + \zeta)^{\frac{p}{2}}$ where $\zeta > 0$ and $0 < p \leq 1$ to approximate the column-sparsity constraint. The $\ell_2/\ell_p$ nonconvex mixed quasi-norm has proven to be a very effective approximator for column sparsity [54]. In addition, as the function is differentiable, it offers an optimization-friendly surrogate.

For the minimum-volume loss, we use $\mathsf{L}_{\mathsf{vol}} = -\log\det(\boldsymbol{F}\boldsymbol{F}^\top)$, where $\boldsymbol{F} = [\boldsymbol{f}(\boldsymbol{x}_1), \ldots, \boldsymbol{f}(\boldsymbol{x}_N)]$. Note that the term encourages maximizing the volume of $\mathrm{conv}\{\boldsymbol{F}\}$, which in turn minimizes the volume of $\mathrm{conv}\{\boldsymbol{W}\}$ under the model $\widetilde{\boldsymbol{G}} = \boldsymbol{W}\boldsymbol{F}$. Therefore, the volume of $\mathrm{conv}\{\boldsymbol{W}\}$ can be minimized via using either $\log\det(\boldsymbol{W}^\top\boldsymbol{W})$ or $-\log\det(\boldsymbol{F}\boldsymbol{F}^\top)$ as the regularizer. The reason why we choose the latter is because minimizing $\log\det(\boldsymbol{W}^\top\boldsymbol{W})$ is ill-defined—the term encourages a rank-deficient $\boldsymbol{W}$ instead of a small-volume full-rank $\boldsymbol{W}$. The remedy in the literature is to use $\log\det(\boldsymbol{W}^\top\boldsymbol{W} + \gamma\boldsymbol{I})$ with $\gamma > 0$ [54] or adding structural constraints (e.g., diagonal dominance) to $\boldsymbol{W}$ [19], but these require more parameter tuning and more prior knowledge.

The constraints on $\boldsymbol{A}_m$'s can be handled by adding `softmax` to the columns. The function class $\mathcal{F}$ can be a designated deep neural network model (e.g., ResNet), where the outputs are also constrained by `softmax`. The constraint on $\boldsymbol{e}_n^{(m)}$ can be handled by parameterizing each $\boldsymbol{e}_n^m = \widetilde{\boldsymbol{e}}_n^{(m)} - (\mathbf{1}^\top \widetilde{\boldsymbol{e}}_n^{(m)})\mathbf{1}/K$, where $\widetilde{\boldsymbol{e}}_n^{(m)}$ is trainable weight vector. More details of the implementation are provided in Appendix G.1. We name our approach as *Crowdsourcing-based Outlier-robust criterion and INstance-dependence-aware deep neural Network learning*, abbreviated as COINNet.

## 4 Related Works

Our development is related to works in transition matrix-based [14–19, 21, 23, 27–29, 33, 47, 55, 56] and sample selection-based [28, 33, 35, 37, 39] noisy label learning, confusion identification in crowdsourcing [14, 18, 57], end-to-end crowdsourcing [20, 30, 31, 49] (with some paying particular attention to incomplete annotations [58, 59]), and identifiability of NMF [53, 60]—see detailed discussions of the related work in Appendix B.

A particular connection to highlight is the usage of multiple annotators as a key enabler of our identifiability guarantees. This result (implicitly) aligns with some recently proposed approaches that advocate for the use of multiple labels in learning under label noise [45, 49]. Specifically, the work by [45] established the identifiability of instance-independent label noise using the consensus of at least three similar data items with the same true label. Similarly, the work by [49] leveraged Kruskal's identifiability-based arguments to recommend using at least three noisy labels to establish the identifiability of the instance-dependent noise transition matrix. These works, particularly [49], were more concerned with the theoretical limitations other than realizable methods. They did not connect the existence of repeated noisy labels to crowdsourcing, but used clusterability properties of datasets to support their arguments. In this work, we advocate *diversity in crowd wisdom*—labels from multiple annotators—based on an intuitive geometric characteristic of the model, which leads to clearly realizable implementations.

# 5 Experiments

**Baselines.** The proposed method is compared with a number of existing baselines in noisy label learning. We choose several end-to-end crowdsourcing methods, namely, `GeoCrowdNet(F)&(W)` [31], `TraceReg` [30], `CrowdLayer` [20], and `Max-MIG` [61]. We also employ four different instance-dependent noisy learning approach, namely, `MEIDTM` [36], `PTD` [27], and `BLTM` [33]—which all use a single annotator. In addition, we present the result of `VolMinNet` [19] that uses the instance-independent confusion model and the volume minimization regularization. We also use two noise-robust loss function-based approaches, namely `GCE` [10] and `Reweight` [21]. For baselines that were developed for single annotator cases, we train them using the labels after majority voting. Note that all the confusion matrix-based noise correction methods (including ours) inherently have a permutation ambiguity (cf. the $\boldsymbol{\Pi}$ term in Theorems 3.5 and 3.6). Hence, we report the highest classification accuracy attained among all possible permutation matrices for every method. In practice, $\boldsymbol{\Pi}$ can be removed by additional annotator inspection on several anchor samples, but we skipped these steps to keep the evaluation simple.

## 5.1 Experiments with Machine Annotations

**Dataset.** We consider the CIFAR-10 [62] and the STL-10 datasets [63]—see Appendix G for details.

**Experiment Setup.** To generate multiple noisy labels, we use $M = 3$ machine annotators for each dataset. In order to simulate a wide range of annotation behaviors, we employ different classification and clustering methods such as *k-nearest neighbors* (kNN), logistic regression, and convolutional neural networks. Each machine annotator is trained by randomly choosing a subset of the training data. We control the labeling accuracies of these machine annotators by varying the size of the dataset and the number of epochs during their training phase. This results in annotators with different labeling qualities, with their average individual noise rates around 20% (*good*), 40% (*medium*), and 70% (*low*). This way, we set up the following three cases: *(i)* High Noise: Three machine annotators are with low, medium, and good quality, respectively. *(i)* Medium Noise: Two machine annotators are with medium quality and one machine annotator with good quality. *(i)* Low Noise: One machine annotators is with medium quality and two machine annotators with good quality. More details are in Appendix G.2.

**Neural Network and Optimizer Settings.** We use ResNet-34 and ResNet-9 architectures [64] as the backbone to run all methods on CIFAR-10 and STL-10 datasets, respectively. For our proposed approach `COINNet`, we fix $\zeta = 10^{-10}, p = 0.4$, and $\mu_1 = \mu_2 = 0.01$. Adam [65] is used as the optimizer with weight decay of $10^{-4}$, learning rate of 0.01, and batch size of 512.

More details and ablation studies with the hyperparameters are provided in Appendix G.

**Results.** Table 1 presents the average testing accuracy of the methods. One can see that our approach `COINNet` consistently outperforms all baselines in different scenarios under test. Notably, the performance gap between `COINNet` and the second-best method is more significant in the high noise regime. For example, in the CIFAR-10 high noise scenario, `COINNet` shows around 4% improvement over the second-best performing baseline `GeoCrowdNet(F)`. Another key observation is that the instance-dependent modeling based baselines such as `MEIDTM`, `PTD`, and `BLTM`, exhibit much degraded performance in high-noise scenarios, possibly due to their multi-stage procedures that accumulate errors easily under such circumstances. In addition, our approach outperforms the instance-independent confusion-based baselines, such as `VolMinNet`, `Reweight` and `GeoCrowdNet(W)`, by nontrivial margins, showing that modeling instance-dependent outliers is indeed beneficial.

## 5.2 Experiments Using Real Annotations

In this section, we use three datasets to evaluate the algorithms.

**CIFAR-10N.** The first dataset that we use is the CIFAR-10N dataset [66]. The data has $N = 60,000$ samples from $K = 10$ classes and the samples were labeled by $M = 3$ anonymous real annotators. The error rates of the 3 annotators are 17.23%, 18.12%, and 17.64%, respectively.

**LabelMe.** We also test the algorithms over the LabelMe dataset [67,68]. The dataset has $N = 2,688$ samples from $K = 8$ classes, and $M = 59$ anonymous real annotators were involved in labeling the data. The average error rate is 25.95%.

Table 1: Average classification accuracy using machine annotators in CIFAR-10 and STL-10 datasets under different labeling scenarios. **Bold black** represents the best and blue represents the second best.

| Method | CIFAR-10 | | | STL-10 | | |
|---|---|---|---|---|---|---|
| | High Noise | Medium Noise | Low Noise | High Noise | Medium Noise | Low Noise |
| PTD | $61.98 \pm 0.35$ | $65.71 \pm 0.69$ | $79.21 \pm 0.41$ | $22.94 \pm 2.15$ | $30.92 \pm 5.45$ | $23.98 \pm 11.52$ |
| BLTM | $52.62 \pm 1.12$ | $61.19 \pm 0.45$ | $71.36 \pm 0.48$ | $31.28 \pm 2.78$ | $30.57 \pm 4.34$ | $32.73 \pm 3.45$ |
| VolMinNet | $49.83 \pm 2.66$ | $50.93 \pm 3.34$ | $70.44 \pm 7.32$ | $42.38 \pm 3.64$ | $60.79 \pm 4.67$ | $64.67 \pm 2.93$ |
| Reweight | $59.73 \pm 1.25$ | $67.57 \pm 0.62$ | $77.40 \pm 0.66$ | $48.57 \pm 8.52$ | $54.23 \pm 3.86$ | $54.67 \pm 6.93$ |
| GCE | $52.14 \pm 0.67$ | $63.90 \pm 0.12$ | $73.26 \pm 0.38$ | $58.35 \pm 0.74$ | $60.65 \pm 0.69$ | $61.70 \pm 2.54$ |
| MEIDTM | $51.38 \pm 0.62$ | $56.20 \pm 0.65$ | $71.14 \pm 0.50$ | $55.50 \pm 0.52$ | $60.20 \pm 0.22$ | $59.84 \pm 3.57$ |
| CrowdLayer | $62.76 \pm 3.12$ | $67.24 \pm 2.26$ | $80.00 \pm 3.02$ | $47.83 \pm 5.51$ | $61.46 \pm 4.58$ | $66.64 \pm 2.48$ |
| TraceReg | $64.90 \pm 0.26$ | $71.21 \pm 0.32$ | $81.81 \pm 0.42$ | $50.14 \pm 5.47$ | $64.50 \pm 0.04$ | $65.67 \pm 3.86$ |
| MaxMIG | $51.92 \pm 1.10$ | $68.45 \pm 0.21$ | $83.16 \pm 0.53$ | $67.92 \pm 0.49$ | $71.75 \pm 0.20$ | $74.32 \pm 0.76$ |
| GeoCrowdNet(F) | $67.22 \pm 0.27$ | $71.91 \pm 0.50$ | $82.18 \pm 0.39$ | $64.70 \pm 0.82$ | $66.15 \pm 0.49$ | $68.40 \pm 0.31$ |
| GeoCrowdNet(W) | $64.08 \pm 0.83$ | $70.01 \pm 0.08$ | $81.53 \pm 0.27$ | $49.30 \pm 4.62$ | $61.54 \pm 4.90$ | $68.37 \pm 0.41$ |
| COINNet (Ours) | $\mathbf{71.22 \pm 0.72}$ | $\mathbf{73.31 \pm 0.09}$ | $\mathbf{84.14 \pm 0.38}$ | $\mathbf{70.12 \pm 0.48}$ | $\mathbf{73.11 \pm 0.37}$ | $\mathbf{76.39 \pm 0.58}$ |

Table 2: Average classification accuracy on CIFAR-10N, LabelMe, and ImageNet-15N datasets, labeled by human annotators. **Bold black** represents the best and blue represents the second best.

| Method/Dataset | CIFAR-10N | LabelMe | ImageNet-15N |
|---|---|---|---|
| PTD | $89.52 \pm 0.24$ | $84.18 \pm 1.36$ | $65.53 \pm 0.18$ |
| BLTM | $75.68 \pm 0.47$ | $82.10 \pm 0.56$ | $66.57 \pm 0.76$ |
| VolMinNet | $86.58 \pm 0.21$ | $79.97 \pm 0.16$ | $63.11 \pm 1.08$ |
| Reweight | $89.56 \pm 0.30$ | $84.51 \pm 0.50$ | $65.85 \pm 2.93$ |
| GCE | $78.01 \pm 7.23$ | $83.41 \pm 0.59$ | $64.71 \pm 1.38$ |
| MEIDTM | $68.69 \pm 0.31$ | $83.53 \pm 0.21$ | $72.66 \pm 0.58$ |
| CrowdLayer | $87.38 \pm 0.43$ | $82.80 \pm 0.90$ | $61.36 \pm 2.73$ |
| TraceReg | $86.57 \pm 0.24$ | $82.83 \pm 0.23$ | $68.43 \pm 0.12$ |
| MaxMIG | $90.11 \pm 0.09$ | $83.73 \pm 0.84$ | $81.13 \pm 1.42$ |
| GeoCrowdNet(F) | $87.19 \pm 0.37$ | $87.21 \pm 0.39$ | $80.45 \pm 1.77$ |
| GeoCrowdNet(W) | $86.43 \pm 0.44$ | $82.83 \pm 0.75$ | $68.79 \pm 0.27$ |
| COINNet (Ours) | $\mathbf{92.09 \pm 0.47}$ | $\mathbf{87.60 \pm 0.54}$ | $\mathbf{93.71 \pm 3.26}$ |

**ImageNet-15N.** In addition to existing datasets, we also acquire noisy annotations by asking AMT workers to annotate some images from ImageNet. We select $K = 15$ classes and submit randomly chosen images to AMT for labeling. Eventually we collect annotations for $N = 2,514$ images from $M = 100$ anonymous real annotators, which serve as our training set. The average error rate of the annotators is 42.68%. The validation and testing sets have 1,462 and 13,157 images, respectively. We release the code and our acquired noisy annotations at `https://github.com/ductri/COINNet`.

**Settings.** For CIFAR-10N, we employ the ResNet-34 architecture to serve as $\boldsymbol{f}$. For the LabelMe and ImageNet-15N datasets, we employ similar settings as in [20]. Specifically, as the training sets are small, the pretrained VGG-16 [69] and CLIP [70] models are used to first extract image embeddings for the experiments on LabelMe and ImageNet-15N, respectively. The embeddings are then fed to $\boldsymbol{f}$, which is a fully connected neural network with one hidden layer and 128 hidden ReLU units. The same encoders and architecture choices are employed for the all methods under test. For our approach COINNet, we set $\mu_1 = \mu_2 = 0.1$ for LabelMe, and $\mu_1 = \mu_2 = 0.01$ for CIFAR-10N and ImageNet-15N.

**Results.** Table 2 presents the average testing accuracy of different methods on the three datasets. The proposed approach, COINNet, shows promising results in all cases, clearly outperforming the baselines by a noticeable edge. This is consistent with the machine annotator experiments.

Fig. 2 demonstrates the outlier identification results using the CIFAR-10N dataset. Here, we define the outlier indicator as $s_n = \sum_{m=1}^{M} \|\widehat{\boldsymbol{e}}_n^{(m)}\|_2, \forall n$, where $\widehat{\boldsymbol{e}}_n^{(m)}$ are the learned instance-dependent perturbations in the model (5). The histogram plot of these scores in Fig. 2 clearly shows that our method does output two types of samples, i.e., nominal samples and outliers, based on these perturbations. In the figure, the images on the middle and the right are examples from the low and high outlier indicator value regimes, respectively. The images with high $s_n$ values show more instance-dependent confusion characteristics (such as background noise and blurring) compared to

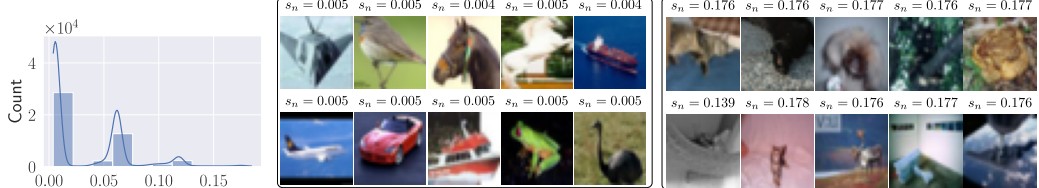

Figure 2: Histogram of the learned outlier indicator values $s_n = \sum_{m=1}^{M} \|\widehat{e}_n^{(m)}\|_2^2$ over all training images in the CIFAR-10N dataset (left), and examples with low (middle) and high (right) $s_n$'s —see more examples in Appendix H.

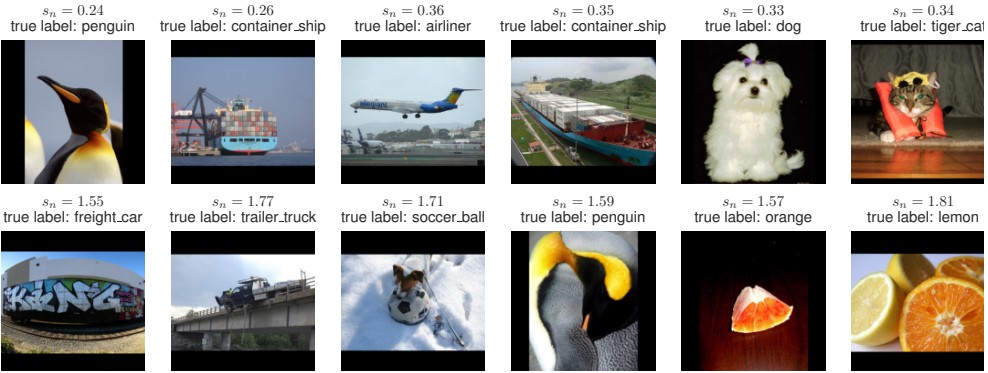

Figure 3: Some examples from ImageNet-15N with low (top) and high (bottom) $s_n$'s

those in the middle. Overall, these results indicate the effectiveness of our outlier-based model in real-world settings.

Fig. 3 shows some examples selected from high-$s_n$ and low-$s_n$ regimes from the results output by `COINNet` in the ImageNet-15N experiment. Similar as before, the images in the first row that have lower $s_n$ scores are visually much easier to recognize. The images in the second row that have about 5 times higher $s_n$ scores are apparently more visually confusing.

More experiments can be found in Appendix H.

## 6   Conclusion

In this work, we considered the noisy label learning problem under a confusion matrix-based model. We developed theory and algorithms in the presence of instance-dependent outliers. Our study revealed that relying solely on single-annotator labels is insufficient for effective outlier detection under the model of interest. We further demonstrated that a crowdsourcing approach, leveraging multiple annotators and a sparsity-constrained loss function, can successfully detect outliers and identify the desired, label noise-free learning system under reasonable conditions. Our analyses and design feature a one-stage differentiable training loss that is optimization-friendly. Empirical results underscore the plausibility of our model and the effectiveness of our proposed method, showing noticeable performance improvements over existing baselines.

**Limitations.** Our work explores a noise transition matrix-based model and treats instance-dependent noisy labels as outliers. The key assumption is that the outliers do not occur overwhelmingly. The assumption is useful, but also could be debatable. In principle, all labels could be generated in an instance-dependent manner. More sophisticated models are needed to deal with this more general case. In addition, our treatment is outlier structure-agnostic. The upshot is that this treatment can also help exclude the negative impacts of other types of outliers. However, the downside is that the structural prior knowledge of the outliers is not fully exploited. If the outlier-generating function could be learned, it can help detect "difficult samples" *before* sending for annotation, which would potentially save resources and reduce annotation errors. We leave this direction for future work.

## Acknowledgment

The work was supported in part by the National Science Foundation (NSF) under Project IIS-2007836.

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

**Supplementary Material of " Noisy Label Learning with Instance-Dependent Outliers: Identifiability via Crowd Wisdom"**

## A    Notation

The notations used in the paper are summarized in Table 3.

Table 3: Definitions of the notations.

| Notation | Definition |
|---|---|
| $x$ | scalar in $\mathbb{R}$ |
| $\boldsymbol{x}$ | vector in $\mathbb{R}^n$, i.e., $\boldsymbol{x} = [x_1, \ldots, x_n]^\top$ |
| $[\boldsymbol{x}]_i$ or $\boldsymbol{x}(k)$ | $i$th entry of the vector $\boldsymbol{x}$ |
| $[\boldsymbol{x}]_{i:j}$ | subvector $\boldsymbol{x}$ such that $[\boldsymbol{x}]_{i:j} = [x_i, x_{i+1}, \ldots, x_j]^\top$, where $j > i$ |
| $\boldsymbol{X}$ | matrix in $\mathbb{R}^{m \times n}$ |
| $[\boldsymbol{X}]_{i,j}$ or $\boldsymbol{X}(i,j)$ | $(i,j)$th entry of $\boldsymbol{X}$ |
| $\boldsymbol{X}(:,i)$ | $i$th column of $\boldsymbol{X}$ |
| $\boldsymbol{X} \geq \boldsymbol{0}$ | $\boldsymbol{X}(i,j) \geq 0, \forall\, (i,j)$ |
| $\boldsymbol{X}[:]$ | vector representing column-wise stacking of $\boldsymbol{X}$ |
| $\|\boldsymbol{x}\|_2$ | Euclidean norm of $\boldsymbol{x}$ |
| $\mathsf{nnz}(\boldsymbol{x})$ | number of non-zero entries of $\boldsymbol{x}$ |
| $\|\boldsymbol{x}\|_0$ | $\ell_0$ norm of $\boldsymbol{x}$; also equal to $\mathsf{nnz}(\boldsymbol{x})$ |
| $\sigma_{\max}(\boldsymbol{X})$ | maximum singular value of $\boldsymbol{X}$ |
| $\sigma_{\min}(\boldsymbol{X})$ | minimum singular value of $\boldsymbol{X}$ |
| $\|\boldsymbol{X}\|_2$ | spectral norm or 2-norm of $\boldsymbol{X}$, $\|\boldsymbol{X}\|_2 = \sigma_{\max}(\boldsymbol{X})$ |
| $\|\boldsymbol{X}\|_\mathrm{F}$ | Frobenius norm of $\boldsymbol{X}$ |
| $\|\boldsymbol{X}\|_{2,1}$ | $\ell_{2,1}$ norm of $\boldsymbol{X}$, i.e., $\|\boldsymbol{X}\|_{2,1} = \sum_{i=1}^n \|\boldsymbol{X}(:,i)\|_2$ |
| $\boldsymbol{\Delta}_K$ | the probability simplex $\{\boldsymbol{x} \in \mathbb{R}^K : \sum_i [\boldsymbol{x}]_i = 1, \boldsymbol{x} \geq \boldsymbol{0}\}$ |
| $\mathsf{cone}(\boldsymbol{X})$ | conic hull of $\boldsymbol{X}$: $\{\boldsymbol{y} \mid \boldsymbol{y} = \boldsymbol{X}\boldsymbol{\theta}, \forall \boldsymbol{\theta} \geq \boldsymbol{0}\}$ |
| $\mathsf{convex}(\boldsymbol{X})$ | convex hull of $\boldsymbol{X}$: $\{\boldsymbol{y} \mid \boldsymbol{y} = \boldsymbol{X}\boldsymbol{\theta}, \forall \boldsymbol{\theta} \geq \boldsymbol{0}, \boldsymbol{1}^\top \boldsymbol{\theta} = 1\}$ |
| $\dagger$ | pseudo-inverse |
| $\top$ | transpose |
| $|\mathcal{C}|$ | the cardinality of the set $\mathcal{C}$ |
| $[T]$ | $\{1, \ldots, T\}$ for an integer $T$ |
| $\boldsymbol{I}_K$ | identity matrix of size $K \times K$ |
| $\boldsymbol{1}$ | all-one vector with proper size |
| $\boldsymbol{0}$ | all-zero vector or matrix with proper size |
| $\overline{\boldsymbol{e}}_i$ | unit vector with the $i$th element being 1 |
| $\mathbb{R}^n_+$ | nonnegative orthant of $\mathbb{R}^n$ |
| $\mathbb{1}[A]$ | Indicator function: $\mathbb{1}[A] = 1$ if the event $A$ happens, otherwise $\mathbb{1}[A] = 0$ |
| $\mathsf{CE}(\boldsymbol{x}, y)$ | cross entropy function: $-\sum_{k=1}^K \mathbb{1}[y = k] \log(\boldsymbol{x}(k))$ |

## B    Related Work

In this section, we briefly review some of the existing works that is relevant to our proposed method. Noisy label learning has been a prominent research focus within the machine learning community for many years—see a survey of these methods in [71]. Among the numerous approaches proposed, those based on label transition matrix-based models (cf. the model in (2)) have been proven effective and several variants of the model have been introduced; see, e.g., [14–19, 21, 23, 27–29, 33, 47, 52, 55, 56]. A broad classification of them is of instance-independent models and instance-dependent models.

**Instance-independent Model-based Methods.**  One of the earliest instance-independent label noise models (i.e., assuming $\boldsymbol{T}^\natural(\boldsymbol{x}_n) = \boldsymbol{A}^\natural, \forall n$ in (2)) is the *random classification model* (RCN) [72–74], where true labels are flipped with a fixed noise rate independently at random. To extend this modeling to more general scenarios, *class-conditioned model* (CCN) was proposed [9], where the noise rates are assumed to be dependent on the true-class the data item belongs. CCN model has led to the

design of several statistically consistent classifiers, as the label transition matrix acts as a correction term for the loss associated with the noisy labels [21, 24, 26, 75]. To provably learn the CCN model, several notions are exploited, e.g., assuming the presence of anchor data items [21], exploiting the clusterability of data items [45], and minimizing the geometric volume of the transition matrix [19].

**Instance-dependent Model-based Approaches.** Instance-dependent label noise models are naturally more plausible. In these models, each data item has a label transition matrix dependent on the data's attributes (cf. Eq. (2)). Recently, a number of methods studied such models by using various simplified representations/parameterizations of the transition matrix $T^\natural(x_n)$, e.g., imposing structural assumptions on label transition matrices [27, 41, 43], bounding noise rates [28], modeling the Bayes label transition matrix or posterior transition matrix instead of the true label transition matrix [33, 37, 40], leveraging additional forms of supervision such as confidence scores [76], and crafting specific regularization terms [36, 39, 42]. To be specific, the work in [27] assumed that the label transition matrix is a convex combination of several instance-independent transition matrices and the combination coefficients depend on the instances. The work in [28] assumed a known bounded noise rate and proposed an instance selection-based strategy. The method proposed in [39] also advocated an instance selection scheme, accompanied with a regularization term to combat instance-dependent label noise. In the work [36], the simplification relies on the assumption that many data items that lie close to each other in the feature space have similar label transition matrices. These interesting developments showed empirical progresses, but the theoretical understanding has been limited. Particularly, the aforementioned identifiability issue was not resolved. In addition, most approaches resort to multi-stage training strategies which may be practically inefficient due to potential error propagation. For example, the approach in [33] first performs instance selection, then estimates the (Bayes-label) transition matrices, and finally estimates the classifier, in a three-stage training scheme. Similar multi-stage approaches are employed in [27, 28, 36].

Our approach is also related to sample-selection-based approaches in noisy label learning, e.g., clean sample selection-based methods [28, 33, 35, 37, 39]. However, most of them treat sample selection as a pre-processing step based on strict prior assumptions (e.g., known noise rates) and then estimate the transition matrix using these selected samples. In contrast, our proposed method and identifiability analysis offer a more principled approach by integrating sample selection with the learning of the classifier and transition matrices in an end-to-end fashion.

## C Proof of Lemma 3.1

Lemma 3.1 is re-stated below:

**Lemma C.1.** *Assume that the observed index pairs $(m, n) \in \mathcal{S}$ are sampled uniformly at random with replacement. Also assume $f^\natural \in \mathcal{F}$, $|\mathcal{I}| \leq C \leq N/2$, and that each $[A_m f(x_n) + e_n^{(m)}]_k$, $\forall A_m \in \mathcal{A}, \forall e_n^{(m)} \in \mathcal{E}, \forall f \in \mathcal{F}$ are lower bounded by $(1/\beta)$ for a certain $\beta > 0$. Then, with probability greater than $1 - \delta$, the following holds:*

$$\frac{1}{NM} \sum_{n=1}^{N} \sum_{m=1}^{M} \|g_n^{\natural(m)} - \widehat{g}_n^{(m)}\|_2^2 \leq \epsilon_g(S), \quad \epsilon_g(S) = \mathcal{O}\left(\beta \mathfrak{R}_\mathcal{S} \log S / \sqrt{S} + \log(\beta) \sqrt{\log(1/\delta)} / \sqrt{S}\right),$$

*where $\mathfrak{R}_\mathcal{S}^2 = K^2 \log(K) + {}^K \|X\|_\mathrm{F}^2 \mathscr{R}_\mathcal{F}/S + CMK \log \sqrt{NM}$ and $X = [x_{n_1}, \ldots, x_{n_S}]$ are the annotated samples, in which $(m_s, n_s) \in \mathcal{S}$ for $s = 1, \ldots, S$.*

The proof is essentially viewing the $g_n^{\natural(m)}$ estimation problem as a quantized matrix completion problem with missing blocks. The key ideas and steps all follow the proof of the end-to-end crowdsourcing work [31, Proposition 1] with the additional consideration of $e_n^{\natural(m)}$. This addition is conceptually not hard to come up with, but the notations can be involved. Thus, we still derive the proof in detail.

**Proof:** Consider the cross entropy-based learning criterion given by

$$\underset{\boldsymbol{A}_m\in\mathcal{A},\boldsymbol{f}\in\mathcal{F},\boldsymbol{e}_n^{(m)}\in\mathcal{E}}{\text{minimize}} \frac{1}{S}\sum_{(m,n)\in\mathcal{S}}\mathsf{CE}(\boldsymbol{A}_m\boldsymbol{f}(\boldsymbol{x}_n)+\boldsymbol{e}_n^{(m)},\widehat{y}_n^{(m)}), \tag{11a}$$

$$\text{subject to } \sum_{n=1}^{N}\mathbb{1}\left\{\sum_{m=1}^{M}\|\boldsymbol{e}_n^{(m)}\|_2>0\right\}\leq C. \tag{11b}$$

where $S=|\mathcal{S}|$, and $\mathsf{CE}(\boldsymbol{x},y)=-\sum_{k=1}^{K}\mathbb{1}[y=k]\log(\boldsymbol{x}(k))$.

Let us denote the objective function in (11a) as follows:

$$\mathsf{L}_{\mathcal{S}}(\boldsymbol{G};\widehat{\mathcal{Y}})\triangleq\frac{1}{S}\sum_{(m,n)\in\mathcal{S}}\mathsf{CE}(\boldsymbol{g}_n^{(m)},\widehat{y}_n^{(m)}), \tag{12}$$

where $\boldsymbol{g}_n^{(m)}$ is given by $\boldsymbol{g}_n^{(m)}=\boldsymbol{A}_m\boldsymbol{f}(\boldsymbol{x}_n)+\boldsymbol{e}_n^{(m)}$, $\boldsymbol{G}$ is given by

$$\boldsymbol{G}=\begin{bmatrix}\boldsymbol{g}_1^{(1)} & \cdots & \boldsymbol{g}_N^{(1)}\\ \vdots & \ddots & \vdots\\ \boldsymbol{g}_1^{(M)} & \cdots & \boldsymbol{g}_N^{(M)}\end{bmatrix}=\begin{bmatrix}\boldsymbol{A}_1\boldsymbol{f}(\boldsymbol{x}_1)+\boldsymbol{e}_1^{(1)} & \cdots & \boldsymbol{A}_1\boldsymbol{f}(\boldsymbol{x}_N)+\boldsymbol{e}_N^{(1)}\\ \vdots & \ddots & \vdots\\ \boldsymbol{A}_M\boldsymbol{f}(\boldsymbol{x}_1)+\boldsymbol{e}_1^{(M)} & \cdots & \boldsymbol{A}_M\boldsymbol{f}(\boldsymbol{x}_N)+\boldsymbol{e}_N^{(M)}\end{bmatrix},$$

and $\widehat{\mathcal{Y}}$ denotes the set of observed noisy labels, i.e., $\{\widehat{y}_n^{(m)}\}_{(m,n)\in\mathcal{S}}$. Let $\{\widehat{\boldsymbol{A}}_m\}$, $\{\widehat{\boldsymbol{e}}_n^{(m)}\}$ and $\widehat{\boldsymbol{f}}$ denote optimal solutions of the learning criterion (11). Let us also define the following:

$$\widehat{\boldsymbol{G}}\triangleq\underset{\boldsymbol{G}\in\mathcal{G}}{\arg\min}\,\mathsf{L}_{\mathcal{S}}(\boldsymbol{G};\widehat{\mathcal{Y}}), \tag{13}$$

where

$$\mathcal{G}\triangleq\{\boldsymbol{G}\in\mathbb{R}^{MK\times N}\mid\boldsymbol{g}_n^{(m)}=\boldsymbol{A}_m\boldsymbol{f}(\boldsymbol{x}_n)+\boldsymbol{e}_n^{(m)},\boldsymbol{f}\in\mathcal{F},\boldsymbol{e}_n^{(m)}\in\mathcal{E},\|\boldsymbol{E}\|_0\leq C,\boldsymbol{A}_m\in\mathcal{A}\},$$

$\mathcal{F}\subset\{\boldsymbol{f}:\mathbb{R}^D\to\mathbb{R}^K\}$ is the neural network function class associated with the classifier function,

$$\mathcal{E}=\{\boldsymbol{e}\in\mathbb{R}^K\mid\boldsymbol{1}^\top\boldsymbol{e}=0\},\quad\|\boldsymbol{E}\|_0=\sum_{n=1}^{N}\mathbb{1}\left\{\sum_{m=1}^{M}\|\boldsymbol{e}_n^{(m)}\|_2>0\right\},$$

$$\mathcal{A}=\{\boldsymbol{A}\in\mathbb{R}^{K\times K}\mid\boldsymbol{1}^\top\boldsymbol{A}=\boldsymbol{1}^\top,\,\boldsymbol{A}\geq\boldsymbol{0}\}.$$

Let $\boldsymbol{G}^\natural$ represents the ground-truth such that

$$\boldsymbol{G}^\natural=\begin{bmatrix}\boldsymbol{g}_1^{\natural(1)} & \cdots & \boldsymbol{g}_N^{\natural(1)}\\ \vdots & \ddots & \vdots\\ \boldsymbol{g}_1^{\natural(M)} & \cdots & \boldsymbol{g}_N^{\natural(M)}\end{bmatrix}=\begin{bmatrix}\boldsymbol{A}_1^\natural\boldsymbol{f}^\natural(\boldsymbol{x}_1)+\boldsymbol{e}_1^{\natural(1)} & \cdots & \boldsymbol{A}_1^\natural\boldsymbol{f}^\natural(\boldsymbol{x}_N)+\boldsymbol{e}_N^{\natural(1)}\\ \vdots & \ddots & \vdots\\ \boldsymbol{A}_M^\natural\boldsymbol{f}^\natural(\boldsymbol{x}_1)+\boldsymbol{e}_1^{\natural(M)} & \cdots & \boldsymbol{A}_M^\natural\boldsymbol{f}^\natural(\boldsymbol{x}_N)+\boldsymbol{e}_N^{\natural(M)}\end{bmatrix}. \tag{14}$$

We aim to bound $\|\boldsymbol{G}^\natural-\widehat{\boldsymbol{G}}\|_\mathrm{F}$. Towards this goal, we adopt the proof strategy based on the viewpoint of matrix completion as employed in [31, 77].

Let us consider that we have $S$ number of observations $\omega_1,\ldots,\omega_S$ such that $\omega_s=(m_s,n_s,\widehat{y}_{n_s}^{(m_s)})$, where each $(m_s,n_s)$ is sampled uniformly at random (with replacement) from $[M]\times[N]$, i.e., the sampling probability for each $(m,n)$ is given by $\pi_{m,n}=\frac{1}{NM}$. In addition, the noisy observations are generated as

$$\widehat{y}_{n_s}^{(m_s)}\sim\mathsf{categorical}(\boldsymbol{g}_{n_s}^{\natural(m_s)}).$$

Let $\mathcal{D}_\omega$ denote the joint probability distribution from where the observations $\omega_s$ are sampled. Then we define the following:

$$\mathsf{L}_{\mathcal{D}}(\boldsymbol{G},\widehat{\mathcal{Y}})\triangleq\mathbb{E}_{\omega\sim\mathcal{D}_\omega}[\mathsf{CE}(\boldsymbol{g}_n^{(m)},\widehat{y}_n^{(m)})].$$

We consider the following set of relations:

$$\mathsf{L}_{\mathcal{D}}(\widehat{\boldsymbol{G}}, \widehat{\mathcal{Y}}) - \mathsf{L}_{\mathcal{D}}(\boldsymbol{G}^{\natural}, \widehat{\mathcal{Y}}) = \mathsf{L}_{\mathcal{D}}(\widehat{\boldsymbol{G}}, \widehat{\mathcal{Y}}) - \mathsf{L}_{\mathcal{S}}(\widehat{\boldsymbol{G}}; \widehat{\mathcal{Y}}) + \mathsf{L}_{\mathcal{S}}(\boldsymbol{G}^{\natural}; \widehat{\mathcal{Y}}) - \mathsf{L}_{\mathcal{D}}(\boldsymbol{G}^{\natural}, \widehat{\mathcal{Y}})$$
$$+ \mathsf{L}_{\mathcal{S}}(\widehat{\boldsymbol{G}}; \widehat{\mathcal{Y}}) - \mathsf{L}_{\mathcal{S}}(\boldsymbol{G}^{\natural}; \widehat{\mathcal{Y}})$$
$$\leq \mathsf{L}_{\mathcal{D}}(\widehat{\boldsymbol{G}}, \widehat{\mathcal{Y}}) - \mathsf{L}_{\mathcal{S}}(\widehat{\boldsymbol{G}}; \widehat{\mathcal{Y}}) + \mathsf{L}_{\mathcal{S}}(\boldsymbol{G}^{\natural}; \widehat{\mathcal{Y}}) - \mathsf{L}_{\mathcal{D}}(\boldsymbol{G}^{\natural}, \widehat{\mathcal{Y}})$$
$$\leq 2 \sup_{\boldsymbol{G} \in \mathcal{G}} \left| \mathsf{L}_{\mathcal{D}}(\boldsymbol{G}, \widehat{\mathcal{Y}}) - \mathsf{L}_{\mathcal{S}}(\boldsymbol{G}; \widehat{\mathcal{Y}}) \right|, \tag{15}$$

where the first inequality is obtained since $\widehat{\boldsymbol{G}}$ is the optimal solution as given by (13) and hence satisfies $\mathsf{L}_{\mathcal{S}}(\widehat{\boldsymbol{G}}; \widehat{\mathcal{Y}}) \leq \mathsf{L}_{\mathcal{S}}(\boldsymbol{G}^{\natural}; \widehat{\mathcal{Y}})$. The last inequality is from the given assumption that $\boldsymbol{G}^{\natural} \in \mathcal{G}$ (This is satisfied since it is given that $\boldsymbol{f}^{\natural} \in \mathcal{F}$ and $\|\boldsymbol{E}^{\natural}\|_0 \leq C$).

Next, we consider the following set of relations:

$$\mathsf{L}_{\mathcal{D}}(\widehat{\boldsymbol{G}}, \widehat{\mathcal{Y}}) - \mathsf{L}_{\mathcal{D}}(\boldsymbol{G}^{\natural}, \widehat{\mathcal{Y}}) = \mathbb{E}_{\omega \sim \mathcal{D}_{\omega}}[\mathsf{CE}(\widehat{\boldsymbol{g}}_n^{(m)}, \widehat{y}_n^{(m)}) - \mathsf{CE}(\boldsymbol{g}_n^{\natural(m)}, \widehat{y}_n^{(m)})]$$

$$= \mathbb{E}_{(m,n,\widehat{y}_n^m) \sim \mathcal{D}_{\omega}}\left[ -\sum_{k=1}^{K} \mathbb{1}[\widehat{y}_n^{(m)} = k] \log[\widehat{\boldsymbol{g}}_n^{(m)}]_k + \sum_{k=1}^{K} \mathbb{1}[\widehat{y}_n^{(m)} = k] \log[\boldsymbol{g}_n^{\natural(m)}]_k \right]$$

$$= \sum_{m=1}^{M} \sum_{n=1}^{N} \pi_{m,n} \mathbb{E}_{\widehat{y}_n^m}\left[ -\sum_{k=1}^{K} \mathbb{1}[\widehat{y}_n^{(m)} = k] \log[\widehat{\boldsymbol{g}}_n^{(m)}]_k + \sum_{k=1}^{K} \mathbb{1}[\widehat{y}_n^{(m)} = k] \log[\boldsymbol{g}_n^{\natural(m)}]_k \right]$$

$$= \frac{1}{NM} \sum_{m=1}^{M} \sum_{n=1}^{N} \left( \sum_{k=1}^{K} -[\boldsymbol{g}_n^{\natural(m)}]_k \log[\widehat{\boldsymbol{g}}_n^{(m)}]_k + \sum_{k=1}^{K} [\boldsymbol{g}_n^{\natural(m)}]_k \log[\boldsymbol{g}_n^{\natural(m)}]_k \right)$$

$$= \frac{1}{NM} \sum_{m=1}^{M} \sum_{n=1}^{N} \sum_{k=1}^{K} [\boldsymbol{g}_n^{\natural(m)}]_k \log \frac{[\boldsymbol{g}_n^{\natural(m)}]_k}{[\widehat{\boldsymbol{g}}_n^{(m)}]_k}$$

$$= \frac{1}{NM} \sum_{m=1}^{M} \sum_{n=1}^{N} \mathsf{D}_{\mathsf{KL}}(\boldsymbol{g}_n^{\natural(m)}, \widehat{\boldsymbol{g}}_n^{(m)})$$

$$\geq \frac{1}{2NM} \sum_{n=1}^{N} \sum_{m=1}^{M} \|\boldsymbol{g}_n^{\natural(m)} - \widehat{\boldsymbol{g}}_n^{(m)}\|_1^2$$

$$\geq \frac{1}{2NM} \sum_{n=1}^{N} \sum_{m=1}^{M} \|\boldsymbol{g}_n^{\natural(m)} - \widehat{\boldsymbol{g}}_n^{(m)}\|_2^2 \tag{16}$$

where we applied Pinsker's inequality [78, 79] for the first inequality and the last inequality uses the fact that $\|\boldsymbol{x}\|_1 \geq \|\boldsymbol{x}\|_2$ for any vector $\boldsymbol{x} \in \mathbb{R}^K$ having $|[\boldsymbol{x}]_k| \leq 1, \forall k \in [K]$. Here $\mathsf{D}_{\mathsf{KL}}(\boldsymbol{g}_1, \boldsymbol{g}_2)$ denotes the Kullback–Leibler divergence between any $\boldsymbol{g}_1, \boldsymbol{g}_2 \in \boldsymbol{\Delta}_K$, i.e., $\mathsf{D}_{\mathsf{KL}}(\boldsymbol{g}_1, \boldsymbol{g}_2) = \sum_{k=1}^{K} [g_1]_k \log \frac{[g_1]_k}{[g_2]_k}$.

Combining (15) with (16), we get

$$\frac{1}{NM} \sum_{n=1}^{N} \sum_{m=1}^{M} \|\boldsymbol{g}_n^{\natural(m)} - \widehat{\boldsymbol{g}}_n^{(m)}\|_2^2 \leq 4 \sup_{\boldsymbol{G} \in \mathcal{G}} \left| \mathsf{L}_{\mathcal{D}}(\boldsymbol{G}, \widehat{\mathcal{Y}}) - \mathsf{L}_{\mathcal{S}}(\boldsymbol{G}; \widehat{\mathcal{Y}}) \right|. \tag{17}$$

Next, we proceed to characterize the R.H.S. of (17). To achieve this, we invoke the following theorem derived from Theorem 26.5 in [80]:

**Theorem C.2.** *[80, Theorem 26.5] Assume that $\forall (m, n, \widehat{y}_n^{(m)})$ and $\forall \boldsymbol{G} \in \mathcal{G}$, we have $|\mathsf{CE}(\boldsymbol{g}_n^{(m)}, \widehat{y}_n^{(m)})| \leq z_{\max}$. Then for any $\boldsymbol{G} \in \mathcal{G}$, the following holds with probability greater than $1 - \delta$:*

$$\left| \mathsf{L}_{\mathcal{D}}(\boldsymbol{G}, \widehat{\mathcal{Y}}) - \mathsf{L}_{\mathcal{S}}(\boldsymbol{G}; \widehat{\mathcal{Y}}) \right| \leq 2\Re(\ell \circ \mathcal{G} \circ \mathcal{S}) + 4z_{\max}\sqrt{\frac{2\log(4/\delta)}{S}}, \tag{18}$$

*where $\ell \circ \mathcal{G} \circ \mathcal{S}$ denotes the set*

$$\ell \circ \mathcal{G} \circ \mathcal{S} \triangleq \left\{ \left( \mathsf{CE}(\boldsymbol{g}_{n_1}^{(m_1)}, \widehat{y}_{n_1}^{(m_1)}), \ldots, \mathsf{CE}(\boldsymbol{g}_{n_S}^{(m_S)}, \widehat{y}_{n_S}^{(m_S)}) \right) \mid \boldsymbol{G} \in \mathcal{G} \right\}$$

*and $\Re(\mathcal{X})$ denotes the empirical Rademacher complexity of the set $\mathcal{X}$.*

To apply Theorem C.2, we need to characterize the terms $z_{\max}$ and $\mathfrak{R}(\ell \circ \mathcal{G} \circ \mathcal{S})$. The term $z_{\max}$ can be characterized as below:

$$z_{\max} = \max_{\substack{[\boldsymbol{g}]_k > \frac{1}{\beta} \\ y \in [K]}} \mathsf{CE}(\boldsymbol{g}, y) \leq \max_{\substack{[\boldsymbol{g}]_k > \frac{1}{\beta} \\ y \in [K]}} -\sum_{k=1}^{K} \mathbb{1}[y = k] \log[\boldsymbol{g}]_k \leq \max_{[\boldsymbol{g}]_k > \frac{1}{\beta}} -\log[\boldsymbol{g}]_k = \log(\beta). \quad (19)$$

Note that in the above relation (19), we utilize the given assumption that each $[\boldsymbol{g}]_k \geq 1/\beta, \forall \boldsymbol{g} \in \mathcal{G}$.

To characterize the Rademacher complexity $\mathfrak{R}(\ell \circ \mathcal{G} \circ \mathcal{S})$, we consider its definition as follows [80]:

$$\mathfrak{R}(\ell \circ \mathcal{G} \circ \mathcal{S}) \triangleq \frac{1}{S} \mathbb{E}\left[\sup_{\boldsymbol{G} \in \mathcal{G}} \sum_{s=1}^{S} \sigma_s \mathsf{CE}(\boldsymbol{g}_{n_s}^{(m_s)}, \widehat{y}_{n_s}^{(m_s)})\right], \quad (20)$$

where expectation is w.r.t. the independent Rademacher random variables $\sigma_s \in \{-1, 1\}$. Since $\boldsymbol{g}_n^{(m)}$ are vectors, we utilize the following result to upper bound $\mathfrak{R}(\ell \circ \mathcal{G} \circ \mathcal{S})$:

**Lemma C.3.** *[81] Assume that the function* $\mathsf{CE}(\cdot, y), \forall y$ *has the Lipschitz constant* $L$. *Then*

$$\mathbb{E}\left[\sup_{\boldsymbol{G} \in \mathcal{G}} \sum_{s=1}^{S} \sigma_s \mathsf{CE}(\boldsymbol{g}_{n_s}^{(m_s)}, \widehat{y}_{n_s}^{(m_s)})\right] \leq \sqrt{2} L \mathbb{E}\left[\sup_{\boldsymbol{G} \in \mathcal{G}} \sum_{s=1}^{S} \sum_{k=1}^{K} \sigma_{sk} [\boldsymbol{g}_{n_s}^{(m_s)}]_k\right],$$

*where* $\sigma_{sk}$ *is an independent (doubly indexed) Rademacher random variable and the expectations are taken w.r.t. the Rademacher random variables.*

To apply Lemma C.3, let us define a vector $\boldsymbol{z} \triangleq \left([\boldsymbol{g}_{n_1}^{(m_1)}]_1, [\boldsymbol{g}_{n_1}^{(m_1)}]_2, \ldots, [\boldsymbol{g}_{n_S}^{(m_S)}]_{K-1}, [\boldsymbol{g}_{n_S}^{(m_S)}]_K\right) \in \mathbb{R}^{SK}$ and the set $\mathcal{Z} \circ \mathcal{S} \triangleq \{\boldsymbol{z} = \left([\boldsymbol{g}_{n_1}^{(m_1)}]_1, [\boldsymbol{g}_{n_1}^{(m_1)}]_2, \ldots, [\boldsymbol{g}_{n_S}^{(m_S)}]_{K-1}, [\boldsymbol{g}_{n_S}^{(m_S)}]_K\right) \mid \boldsymbol{G} \in \mathcal{G}\}$. Using these definitions, let us apply Lemma C.3 in (20) and get the following relation:

$$\begin{aligned} \mathfrak{R}(\ell \circ \mathcal{G} \circ \mathcal{S}) &\leq \frac{\sqrt{2}\beta}{S} \mathbb{E}\left[\sup_{\boldsymbol{G} \in \mathcal{G}} \sum_{s=1}^{S} \sum_{k=1}^{K} \sigma_{sk} [\boldsymbol{g}_{n_s}^{(m_s)}]_k\right] \\ &= \frac{\sqrt{2}\beta}{S} \mathbb{E}\left[\sup_{\boldsymbol{z} \in \mathcal{Z} \circ \mathcal{S}} \sum_{i}^{SK} \sigma_i [\boldsymbol{z}]_i\right] \\ &= \sqrt{2}\beta K \frac{1}{SK} \mathbb{E}\left[\sup_{\boldsymbol{z} \in \mathcal{Z} \circ \mathcal{S}} \sum_{i=1}^{SK} \sigma_i [\boldsymbol{z}]_i\right] = \sqrt{2}\beta K \mathfrak{R}(\mathcal{Z} \circ \mathcal{S}), \quad (21) \end{aligned}$$

where $\beta$ is an upper bound of the Lipschitz constant of the cross entropy loss function $\mathsf{CE}(\boldsymbol{g}, y) = -\sum_{k=1}^{K} \mathbb{1}[y = k] \log[\boldsymbol{g}]_k$ when $\boldsymbol{g} \in \boldsymbol{\Delta}_K$ with $[\boldsymbol{x}]_k > (1/\beta). \forall k$.

Next, we will characterize $\mathfrak{R}(\mathcal{Z} \circ \mathcal{S})$ using the covering number of the set $\mathcal{Z} \circ \mathcal{S}$. To define the covering number, let us first consider the concept of $\epsilon$-net covering of a set. The $\epsilon$-net covering of a set $\mathcal{X}$ (denoted as $\overline{\mathcal{X}}_\epsilon$) is defined as the finite subset of $\mathcal{X}$ (i.e., $\overline{\mathcal{X}}_\epsilon \subseteq \mathcal{X}$) such that for any $\boldsymbol{x} \in \mathcal{X}$, there exists an $\overline{\boldsymbol{x}} \in \overline{\mathcal{X}}_\epsilon$ satisfying $\|\boldsymbol{x} - \overline{\boldsymbol{x}}\|_2 \leq \epsilon$ [82]. The smallest cardinality of the $\epsilon$-nets of $\mathcal{X}$ is known as the covering number of $\mathcal{X}$, which is denoted as $\overline{\mathsf{N}}(\epsilon, \mathcal{X})$ .

Let us consider a pair of vectors $\boldsymbol{z}_\mathcal{S}, \overline{\boldsymbol{z}}_\mathcal{S} \in \mathcal{Z} \circ \mathcal{S}$ as below:

$$\boldsymbol{z}_\mathcal{S} = \left([\boldsymbol{g}_{n_1}^{(m_1)}]_1, [\boldsymbol{g}_{n_1}^{(m_1)}]_2, \ldots, [\boldsymbol{g}_{n_S}^{(m_S)}]_{K-1}, [\boldsymbol{g}_{n_S}^{(m_S)}]_K\right) \in \mathbb{R}^{SK}, \quad \boldsymbol{g}_{n_s}^{(m_s)} = \boldsymbol{A}_{m_s} \boldsymbol{f}(\boldsymbol{x}_{n_s}) + \boldsymbol{e}_{n_s}^{(m_s)},$$

$$\overline{\boldsymbol{z}}_\mathcal{S} = \left([\overline{\boldsymbol{g}}_{n_1}^{(m_1)}]_1, [\overline{\boldsymbol{g}}_{n_1}^{(m_1)}]_2, \ldots, [\overline{\boldsymbol{g}}_{n_S}^{(m_S)}]_{K-1}, [\overline{\boldsymbol{g}}_{n_S}^{(m_S)}]_K\right) \in \mathbb{R}^{SK}, \quad \overline{\boldsymbol{g}}_{n_s}^{(m_s)} = \overline{\boldsymbol{A}}_{m_s} \overline{\boldsymbol{f}}(\boldsymbol{x}_{n_s}) + \overline{\boldsymbol{e}}_{n_s}^{(m_s)},$$

where $\overline{\boldsymbol{A}}_{m_s}$ belongs to the $\epsilon$-net covering of $\mathcal{A}$, $\overline{\boldsymbol{f}}$ belongs to the $\epsilon$-net covering of $\mathcal{F}$, and $\overline{\boldsymbol{e}}_{n_s}^{(m_s)}$ is the appropriate sub-vector belonging to the $\epsilon$-net covering of $\mathcal{E}_0$.

Let us also consider the following definitions:

$$\boldsymbol{u}_{\mathcal{S}} = ([\boldsymbol{A}_{m_1}\boldsymbol{f}(\boldsymbol{x}_{n_1})]_1, [\boldsymbol{A}_{m_1}\boldsymbol{f}(\boldsymbol{x}_{n_1})]_2, \ldots, [\boldsymbol{A}_{m_S}\boldsymbol{f}(\boldsymbol{x}_{n_S})]_{K-1}, [\boldsymbol{A}_{m_S}\boldsymbol{f}(\boldsymbol{x}_{n_S})]_K) \in \mathbb{R}^{SK},$$

$$\overline{\boldsymbol{u}}_{\mathcal{S}} = ([\overline{\boldsymbol{A}}_{m_1}\overline{\boldsymbol{f}}(\boldsymbol{x}_{n_1})]_1, [\overline{\boldsymbol{A}}_{m_1}\overline{\boldsymbol{f}}(\boldsymbol{x}_{n_1})]_2, \ldots, [\overline{\boldsymbol{A}}_{m_S}\overline{\boldsymbol{f}}(\boldsymbol{x}_{n_S})]_{K-1}, [\overline{\boldsymbol{A}}_{m_S}\overline{\boldsymbol{f}}(\boldsymbol{x}_{n_S})]_K) \in \mathbb{R}^{SK},$$

$$\boldsymbol{e}_{\mathcal{S}} = \left([\boldsymbol{e}_{n_1}^{(m_1)}]_1, [\boldsymbol{e}_{n_1}^{(m_1)}]_2, \ldots, [\boldsymbol{e}_{n_S}^{(m_S)}]_{K-1}, [\boldsymbol{e}_{n_S}^{(m_S)}]_K\right) \in \mathbb{R}^{SK},$$

$$\overline{\boldsymbol{e}}_{\mathcal{S}} = \left([\overline{\boldsymbol{e}}_{n_1}^{(m_1)}]_1, [\overline{\boldsymbol{e}}_{n_1}^{(m_1)}]_2, \ldots, [\overline{\boldsymbol{e}}_{n_S}^{(m_S)}]_{K-1}, [\overline{\boldsymbol{e}}_{n_S}^{(m_S)}]_K\right) \in \mathbb{R}^{SK},$$

$$\boldsymbol{e}_{\mathcal{N}\mathcal{M}} = \left([\boldsymbol{e}_1^{(1)}]_1, [\boldsymbol{e}_1^{(1)}]_2, \ldots, [\boldsymbol{e}_N^{(M)}]_{K-1}, [\boldsymbol{e}_N^{(M)}]_K\right) \in \mathbb{R}^{NMK},$$

$$\overline{\boldsymbol{e}}_{\mathcal{N}\mathcal{M}} = \left([\overline{\boldsymbol{e}}_1^{(1)}]_1, [\overline{\boldsymbol{e}}_1^{(1)}]_2, \ldots, [\overline{\boldsymbol{e}}_N^{(M)}]_{K-1}, [\overline{\boldsymbol{e}}_N^{(M)}]_K\right) \in \mathbb{R}^{NMK},$$

Then, we have the following relation due to triangle inequality

$$\|\boldsymbol{z}_{\mathcal{S}} - \overline{\boldsymbol{z}}_{\mathcal{S}}\|_2 \le \|\boldsymbol{u}_{\mathcal{S}} - \overline{\boldsymbol{u}}_{\mathcal{S}}\|_2 + \|\boldsymbol{e}_{\mathcal{S}} - \overline{\boldsymbol{e}}_{\mathcal{S}}\|_2. \tag{22}$$

Next, consider the second term on the R.H.S. of (22):

$$\|\boldsymbol{e}_{\mathcal{S}} - \overline{\boldsymbol{e}}_{\mathcal{S}}\|_2^2 = \sum_{s=1}^{S} \|\boldsymbol{e}_{n_s}^{(m_s)} - \overline{\boldsymbol{e}}_{n_s}^{(m_s)}\|_2^2 \le S\|\boldsymbol{e}_{\mathcal{N}\mathcal{M}} - \overline{\boldsymbol{e}}_{\mathcal{N}\mathcal{M}}\|_2^2. \tag{23}$$

We also have the following relations for the first term on the R.H.S. of (22) :

$$\begin{aligned}
\|\boldsymbol{u}_{\mathcal{S}} - \overline{\boldsymbol{u}}_{\mathcal{S}}\|^2 &= \sum_{s=1}^{S} \|\boldsymbol{A}_{m_s}\boldsymbol{f}(\boldsymbol{x}_{n_s}) - \overline{\boldsymbol{A}}_{m_s}\overline{\boldsymbol{f}}(\boldsymbol{x}_{n_s})\|_2^2 \\
&= \sum_{s=1}^{S} \|\boldsymbol{A}_{m_s}\boldsymbol{f}(\boldsymbol{x}_{n_s}) - \overline{\boldsymbol{A}}_{m_s}\boldsymbol{f}(\boldsymbol{x}_{n_s}) + \overline{\boldsymbol{A}}_{m_s}\boldsymbol{f}(\boldsymbol{x}_{n_s}) - \overline{\boldsymbol{A}}_{m_s}\overline{\boldsymbol{f}}(\boldsymbol{x}_{n_s})\|_2^2 \\
&\le \sum_{s=1}^{S} \left(\|\boldsymbol{A}_{m_s} - \overline{\boldsymbol{A}}_{m_s}\|_{\mathrm{F}}\|\boldsymbol{f}(\boldsymbol{x}_{n_s})\|_2 + \|\overline{\boldsymbol{A}}_{m_s}\|_{\mathrm{F}}\|\boldsymbol{f}(\boldsymbol{x}_{n_s}) - \overline{\boldsymbol{f}}(\boldsymbol{x}_{n_s})\|\right)^2 \\
&\le 2\sum_{s=1}^{S} \|\boldsymbol{A}_{m_s} - \overline{\boldsymbol{A}}_{m_s}\|_{\mathrm{F}}^2\|\boldsymbol{f}(\boldsymbol{x}_{n_s})\|_2^2 + 2\sum_{s=1}^{S}\|\overline{\boldsymbol{A}}_{m_s}\|_{\mathrm{F}}^2\|\boldsymbol{f}(\boldsymbol{x}_{n_s}) - \overline{\boldsymbol{f}}(\boldsymbol{x}_{n_s})\|^2 \\
&\le 2\sum_{s=1}^{S} \|\boldsymbol{A}_{m_s} - \overline{\boldsymbol{A}}_{m_s}\|_{\mathrm{F}}^2 + 2K\sum_{s=1}^{S}\|\boldsymbol{f}(\boldsymbol{x}_{n_s}) - \overline{\boldsymbol{f}}(\boldsymbol{x}_{n_s})\|^2. \tag{24}
\end{aligned}$$

In the above set of relations, the first inequality is by triangle inequality. The second inequality uses that fact that $(a+b)^2 \le 2a^2 + 2b^2$. The third inequality uses the fact that the Frobenius norm of $\overline{\boldsymbol{A}}_m(\boldsymbol{x}_n)$'s are bounded by $\sqrt{K}$ and the $\ell_2$ norm of $\boldsymbol{f}(\boldsymbol{x}_n)$ is bounded by 1. Hence, combining (22)-(24), we get that in order to obtain an $\varepsilon$-net covering for the set $\mathcal{Z} \circ \mathcal{S}$ (i.e., $\|\boldsymbol{z} - \overline{\boldsymbol{z}}\|_2 \le \varepsilon$), we only need to show that there exists a $\frac{\varepsilon}{4\sqrt{K}}$-net covering for $\mathcal{F} \circ \mathcal{S}$, an $\frac{\varepsilon}{2\sqrt{S}}$-net covering for $\mathcal{E}_{\mathcal{N}\mathcal{M}}$, $\forall m$, and a $\frac{\varepsilon}{4\sqrt{S}}$-net covering for each $\mathcal{A}$'s. Here we have

$$\mathcal{F} \circ \mathcal{S} = \{[\boldsymbol{f}(\boldsymbol{x}_1), \ldots, \boldsymbol{f}(\boldsymbol{x}_S)] \in \mathbb{R}^{K \times S} \mid \boldsymbol{f} \in \mathcal{F}, (m,n) \in \mathcal{S}\},$$

$$\mathcal{E}_{\mathcal{N}\mathcal{M}} = \{[\boldsymbol{e}_1^{(1)\top}, \ldots, \boldsymbol{e}_N^{(M)\top}]^\top \in \mathbb{R}^{KNM} \mid \boldsymbol{e}_n^{(m)} \in \mathcal{E}, \|\boldsymbol{E}\|_0 \le C, \},$$

$\|\boldsymbol{E}\|_0 = \sum_{n=1}^{N} \mathbb{1}\left\{\sum_{m=1}^{M} \|\boldsymbol{e}_n^{(m)}\|_2 > 0\right\}$. Note that the full rank matrix $\boldsymbol{A}_m \in \mathbb{R}^{K \times K}$ can be represented as a $K^2$-dimensional vector whose Euclidean norm is bounded by $\sqrt{K}$. Hence, the cardinality of the $\frac{\varepsilon}{4\sqrt{S}}$-net covering for $\boldsymbol{A}_m \in \mathbb{R}^{K \times K}$ is at most $\left(\frac{8K\sqrt{SK}}{\varepsilon}\right)^{K^2}$ [80]. We consider the covering number corresponding to the function class $\mathcal{F} \circ \mathcal{S}$. Towards this, we invoke the following lemma:

**Lemma C.4.** *[51, Theorem 3.3] Let fixed nonlinearities $(\sigma_1, \ldots, \sigma_L)$ and reference matrices $(M_1, \ldots, M_L)$ be given, where $\sigma_i$ is $\rho_i$-Lipschitz and $\sigma_i(0) = 0$. Let spectral norm bounds $(s_1, \ldots, s_L)$, and matrix (2,1)-norm bounds $b_1, \ldots, b_L$ be given. Let $\boldsymbol{x}_1, \ldots, \boldsymbol{x}_S$ be the S number of data items. Let $\boldsymbol{h}$ belongs to the following function class:*

$$\mathcal{H} \triangleq \{\boldsymbol{h} : \mathbb{R}^D \to \mathbb{R}^K \mid \boldsymbol{h}(\boldsymbol{x}) = \sigma_L(\boldsymbol{C}_1 \sigma_{L-1}(\boldsymbol{C}_{L-1} \ldots \sigma_1(\boldsymbol{C}_1(\boldsymbol{x}) \ldots))), \forall x \in \mathbb{R}^D\},$$

*where $\boldsymbol{C}_i \in \mathbb{R}^{d_i \times d_{i-1}}$ with $d_0 = D$ and $d_L = K$. Let $\mathcal{H} \circ \mathcal{S}$ denote the family of matrices obtained by evaluating the data items with all choices of network $\mathcal{H}$:*

$$\mathcal{H} \circ \mathcal{S} \triangleq \{[\boldsymbol{h}(\boldsymbol{x}_1), \ldots, \boldsymbol{h}(\boldsymbol{x}_S)] \mid \mathcal{C} = (\boldsymbol{C}_1, \ldots, \boldsymbol{C}_L), \|\boldsymbol{C}_i\|_2 \leq s_i, \|\boldsymbol{C}_i - \boldsymbol{M}_i\|_{2,1} \leq b_i, \boldsymbol{h} \in \mathcal{H}\},$$

*Then for any $\epsilon > 0$,*

$$\log \overline{\mathsf{N}}(\epsilon, \mathcal{H} \circ \mathcal{S}) \leq \frac{\|\boldsymbol{X}\|_{\mathrm{F}}^2 \mathscr{R}_{\mathcal{H}}}{\epsilon^2},$$

*where $\boldsymbol{X} = [\boldsymbol{x}_1, \ldots, \boldsymbol{x}_S]$, $\mathscr{R}_{\mathcal{H}} = \log(2H^2)\left(\prod_{j=1}^L s_j^2 \rho_j^2\right)\left(\sum_L^i (b_i/s_i)^{2/3}\right)^3$ is called as the spectral-complexity upper bound of neural network function class $\mathcal{H}$ and $H = \max_\ell d_\ell$.*

Using Lemma C.4, we get the cardinality of the $\frac{\varepsilon}{8\sqrt{SK}}$-net covering for $\mathcal{F} \circ \mathcal{N}$ as below:

$$\overline{\mathsf{N}}\left(\frac{\varepsilon}{8\sqrt{SK}}, \mathcal{F} \circ \mathcal{S}\right) \leq \exp\left(\frac{16K\|\boldsymbol{X}_\mathcal{S}\|_{\mathrm{F}}^2 \mathscr{R}_{\mathcal{F}}}{\varepsilon^2}\right),$$

where $\boldsymbol{X}_\mathcal{S} = [\boldsymbol{x}_1, \ldots, \boldsymbol{x}_S] \in \mathbb{R}^{d \times S}$ and the parameter $\mathscr{R}_{\mathcal{F}}$ the spectral-complexity upper bound of $\mathcal{F}$.

Next, we proceed to get the cardinality of the $\frac{\varepsilon}{2\sqrt{S}}$-net covering for $\mathcal{E}_{\mathcal{NM}}$. Here, $\mathcal{E}_{\mathcal{NM}}$ is a set of sparse vectors with the number of non-zero elements upper-bounded by $CMK$, i.e., for any $e_{\mathcal{NM}} \in \mathcal{E}_{\mathcal{NM}}, \|e_{\mathcal{NM}}\|_0 \leq CMK$. To get the covering number of a set of sparse vectors, we invoke the following result:

**Lemma C.5.** *[83] Let $\mathcal{X} = \{\boldsymbol{z} \in \mathbb{R}^N \mid \|\boldsymbol{z}\|_2 \leq q, \|\boldsymbol{z}\|_0 \leq C\}$. Suppose that $C \leq \frac{N}{2}$. Then, there exists an $\epsilon$-net covering denoted as $\overline{\mathcal{X}}_\epsilon$ such that*

$$\overline{\mathsf{N}}(\epsilon, \mathcal{X}) \leq \left(\frac{eN}{C} \frac{q}{\epsilon}\right)^C,$$

*where $e$ is the Euler's number.*

Applying Lemma C.5, we get the cardinality of $\frac{\varepsilon}{2\sqrt{S}}$-net covering of of $\mathcal{E}_{\mathcal{NM}}$ as follows:

$$\overline{\mathsf{N}}\left(\frac{\varepsilon}{2\sqrt{S}}, \mathcal{E}_{\mathcal{NM}}\right) \leq \left(\frac{eNMK}{CMK} \frac{4CM\sqrt{S}}{\varepsilon}\right)^{CMK}$$

$$= \left(\frac{4eNM\sqrt{S}}{\varepsilon}\right)^{CMK}. \tag{25}$$

Note that in (25), we used the result that $\|e_{\mathcal{NM}}\|_2 \leq \sqrt{2CM} \leq 2CM, \forall e_{\mathcal{NM}} \in \mathcal{E}_{\mathcal{NM}}$.

Using the covering number results, the cardinality of the $\varepsilon$-net covering of set $\mathcal{Z} \circ \mathcal{S}$ is bounded by the following:

$$\overline{\mathsf{N}}(\varepsilon, \mathcal{Z} \circ \mathcal{S}) \leq \exp\left(\frac{16K\|\boldsymbol{X}_\mathcal{S}\|_{\mathrm{F}}^2 \mathcal{R}_{\mathcal{F}}}{\varepsilon^2}\right) \times \left(\frac{4eNM\sqrt{S}}{\varepsilon}\right)^{CMK} \times \left(\frac{8K\sqrt{SK}}{\varepsilon}\right)^{K^2}$$

$$= \exp\left(\frac{16K\|\boldsymbol{X}_\mathcal{S}\|_{\mathrm{F}}^2 \mathscr{R}_{\mathcal{F}}}{\varepsilon^2} + CMK \log \frac{4eNM\sqrt{S}}{\varepsilon} + K^2 \log\left(\frac{8K\sqrt{SK}}{\varepsilon}\right)\right). \tag{26}$$

Now that we have characterized $\overline{\mathsf{N}}(\varepsilon, \mathcal{Z} \circ \mathcal{S})$, we invoke the Dudley entropy chaining technique lemma to obtain the Rademacher complexity $\mathfrak{R}(\mathcal{Z} \circ \mathcal{S})$ [80]:

**Lemma C.6.** *Consider set $A \subseteq \mathbb{R}^m$, let $c = \min_{\overline{a}} \max_{a \in A} \|\overline{a} - a\|$. Then for any integer $T > 0$,*

$$\mathfrak{R}(A) \leq \frac{c2^{-T}}{\sqrt{m}} + \frac{6c}{m} \sum_{t=1}^{T} 2^{-t} \sqrt{\log \overline{\mathsf{N}}(c2^{-t}, A)}.$$

Since,

$$\|z\| \leq \sqrt{S}, \quad \forall z \in \mathcal{Z} \circ \mathcal{S},$$

applying Lemma C.6 on set $\mathcal{Z} \circ \mathcal{S}$ gives

$$\mathfrak{R}(\mathcal{Z} \circ \mathcal{S})$$

$$\leq \frac{\sqrt{S}2^{-T}}{\sqrt{SK}} + \frac{6\sqrt{S}}{SK} \sum_{t=1}^{T} 2^{-t} \sqrt{\log \overline{\mathsf{N}}(\sqrt{S}2^{-t}, \mathcal{Z} \circ \mathcal{S})}$$

$$\leq \frac{2^{-T}}{\sqrt{K}} + \frac{6}{K\sqrt{S}} \sum_{t=1}^{T} 2^{-t} \sqrt{\frac{16K\|\boldsymbol{X}_{\mathcal{S}}\|_{\mathrm{F}}^2 \mathscr{R}_{\mathcal{F}}}{S4^{-t}} + CMK \log \frac{4eNM}{2^{-t}} + K^2 \log \left( \frac{8K\sqrt{K}}{2^{-t}} \right)}$$

$$\leq \frac{2^{-T}}{\sqrt{K}} + \frac{6}{K\sqrt{S}} \sum_{t=1}^{T} \sqrt{\frac{16K\|\boldsymbol{X}_{\mathcal{S}}\|_{\mathrm{F}}^2 \mathscr{R}_{\mathcal{F}}}{S} + 4^{-t} \left( CMK \log \frac{4eNM}{2^{-t}} + K^2 \log \left( \frac{8K\sqrt{K}}{2^{-t}} \right) \right)}$$

$$\leq \frac{2^{-T}}{\sqrt{K}} + \frac{6}{K\sqrt{S}} \sum_{t=1}^{T} \sqrt{\frac{16K\|\boldsymbol{X}_{\mathcal{S}}\|_{\mathrm{F}}^2 \mathscr{R}_{\mathcal{F}}}{S} + CMK \log(4eNM) + K^2 \log\left(8K\sqrt{K}\right)}$$

$$= \frac{2^{-T}}{\sqrt{K}} + \frac{6T}{K\sqrt{S}} \sqrt{\frac{16K\|\boldsymbol{X}_{\mathcal{S}}\|_{\mathrm{F}}^2 \mathscr{R}_{\mathcal{F}}}{S} + CMK \log(4eNM) + K^2 \log\left(8K\sqrt{K}\right)}.$$

Choosing $T = \log S / (2 \log 2)$, we get the bound

$$\mathfrak{R}(\mathcal{Z} \circ \mathcal{S}) \leq \frac{1}{\sqrt{SK}}$$

$$+ \frac{6 \log S}{2K\sqrt{S} \log 2} \sqrt{\frac{16K\|\boldsymbol{X}_{\mathcal{S}}\|_{\mathrm{F}}^2 \mathscr{R}_{\mathcal{F}}}{S} + CMK \log(4eNM) + K^2 \log\left(8K\sqrt{K}\right)}. \qquad (27)$$

Combining the upper bound of $\mathfrak{R}(\mathcal{Z} \circ \mathcal{S})$ given by (27) with the upper bound of $\mathfrak{R}(\ell \circ \mathcal{G} \circ \mathcal{S})$ as given by (21) and with the results in (18) and (19), we get that with probability greater than $1 - \delta$,

$$\left| \mathsf{L}_{\mathcal{D}}(\boldsymbol{G}, \widehat{\mathcal{Y}}) - \mathsf{L}_{\mathcal{S}}(\boldsymbol{G}; \widehat{\mathcal{Y}}) \right| \leq 2\sqrt{2}\beta K \mathfrak{R}(\mathcal{Z} \circ \mathcal{S}) + 4 \log(\beta) \sqrt{\frac{2 \log(4/\delta)}{S}}, \qquad (28)$$

where $\mathfrak{R}_{\mathcal{S}}(\mathcal{Z} \circ \mathcal{S})$ is upper bounded by (27).

The above relation, combined with (16), implies that with probability greater than $1 - 2\delta$:

$$\frac{1}{NM} \sum_{n=1}^{N} \sum_{m=1}^{M} \|\boldsymbol{g}_n^{\natural(m)} - \widehat{\boldsymbol{g}}_n^{(m)}\|_2^2 \leq 4\sqrt{2}\beta K \mathfrak{R}(\mathcal{Z} \circ \mathcal{S}) + 8 \log(\beta) \sqrt{\frac{2 \log(4/\delta)}{S}}, \qquad (29)$$

where $\mathfrak{R}(\mathcal{Z} \circ \mathcal{S})$ is upper-bounded in (27).

## D  Proof of Theorem 3.5

To obtain the results in Theorem 3.5, we first show that the criterion correctly distinguishes the instance-dependent and instance-independent samples, i.e., $\widehat{\mathcal{I}} = \mathcal{I}$ where we define $\widehat{\mathcal{I}} \triangleq \{i \in [N] \mid \widehat{e}_i \neq \boldsymbol{0}\}$.

Recall $\widehat{\boldsymbol{A}}_m$'s, $\widehat{\boldsymbol{e}}_n^{(m)}$'s and $\widehat{\boldsymbol{f}}$ are defined as in Theorem 3.5, and let $\widehat{\boldsymbol{e}}_n = [(\widehat{\boldsymbol{e}}_n^{(1)})^\top, \ldots, (\widehat{\boldsymbol{e}}_n^{(M)})^\top]^\top$. We first show that $\widehat{\mathcal{I}}^c \subseteq \mathcal{I}^c$. From Lemma 3.1, the following holds:

$$\|\boldsymbol{W}^\natural \boldsymbol{F}^\natural + \boldsymbol{E}^\natural - \widehat{\boldsymbol{W}}\widehat{\boldsymbol{F}} - \widehat{\boldsymbol{E}}\|_{\mathrm{F}} \leq \epsilon_{\boldsymbol{g}}(S) \qquad (30)$$

Then the inequality also holds for any submatrix of the left hand side matrix. In particular, for $i \in \mathcal{I}$ and $S > S_0$,

$$\epsilon_{\boldsymbol{g}}(S_0) > \epsilon_{\boldsymbol{g}}(S)$$

$$\geq \|\boldsymbol{W}^{\natural}\boldsymbol{F}^{\natural}(:,\{i\}\cap\mathcal{I}^c\cap\widehat{\mathcal{I}}^c) + \boldsymbol{E}^{\natural}(:,\{i\}\cap\mathcal{I}^c\cap\widehat{\mathcal{I}}^c) - \widehat{\boldsymbol{W}}\widehat{\boldsymbol{F}}(:,\{i\}\cap\mathcal{I}^c\cap\widehat{\mathcal{I}}^c) - \widehat{\boldsymbol{E}}(:,\{i\}\cap\mathcal{I}^c\cap\widehat{\mathcal{I}}^c)\|_{\mathrm{F}}$$

$$= \|\boldsymbol{W}^{\natural}[\boldsymbol{f}_i^{\natural}, \boldsymbol{F}^{\natural}(:,\mathcal{I}^c\cap\widehat{\mathcal{I}}^c)] + [\boldsymbol{e}_i^{\natural},\boldsymbol{0},\ldots,\boldsymbol{0}] - \widehat{\boldsymbol{W}}[\widehat{\boldsymbol{f}}_i,\widehat{\boldsymbol{F}}(:,\mathcal{I}^c\cap\widehat{\mathcal{I}}^c)] - [\widehat{\boldsymbol{e}}_i,\boldsymbol{0},\ldots,\boldsymbol{0}]\|_{\mathrm{F}}$$

$$\geq \|\boldsymbol{W}^{\natural}[\boldsymbol{f}_i^{\natural}, \boldsymbol{F}^{\natural}(:,\mathcal{I}^c\cap\widehat{\mathcal{I}}^c)] + [\boldsymbol{e}_i^{\natural},\boldsymbol{0},\ldots,\boldsymbol{0}] - \widehat{\boldsymbol{W}}[\widehat{\boldsymbol{f}}_i,\widehat{\boldsymbol{F}}(:,\mathcal{I}^c\cap\widehat{\mathcal{I}}^c)]\|_{\mathrm{F}} - \|[\widehat{\boldsymbol{e}}_i,\boldsymbol{0},\ldots,\boldsymbol{0}]\|_{\mathrm{F}}$$

$$\geq \kappa(\boldsymbol{e}_i^{\natural}) - \|[\widehat{\boldsymbol{e}}_i,\boldsymbol{0},...,\boldsymbol{0}]\|_{\mathrm{F}} \quad \text{(by Definition 3.3)}$$

$$\geq \epsilon_{\boldsymbol{g}}(S_0) - \|[\widehat{\boldsymbol{e}}_i,\boldsymbol{0},...,\boldsymbol{0}]\|_{\mathrm{F}} \quad \text{(by the assumption in Theorem 3.5).}$$

Therefore, $\widehat{\boldsymbol{e}}_i \neq \boldsymbol{0}$ and that concludes $\widehat{\mathcal{I}}^c \subseteq \mathcal{I}$. On the other hand, since $\widehat{\boldsymbol{e}}_n^{(m)}$'s minimizes $\sum_{n=1}^N \mathbb{1}\left\{\sum_{m=1}^M \|\widehat{\boldsymbol{e}}_n^{(m)}\|_2 > 0\right\}$, it holds that $\mathcal{I}^c \subseteq \widehat{\mathcal{I}}^c$. This implies that $\mathcal{I}^c = \widehat{\mathcal{I}}^c$ must hold.

With $\mathcal{I}^c = \widehat{\mathcal{I}}^c$, we have the following relation from (30):

$$\frac{1}{MN}\|\boldsymbol{W}^{\natural}\boldsymbol{F}^{\natural}(:,\mathcal{I}^c) - \widehat{\boldsymbol{W}}\widehat{\boldsymbol{F}}(:,\mathcal{I}^c)\|_{\mathrm{F}}^2 \leq 4\sqrt{2}\beta K \mathfrak{R}(\mathcal{Z}\circ\mathcal{S}) + 8\log(\beta)\sqrt{\frac{2\log(4/\delta)}{S}}, \qquad (31)$$

To proceed, we invoke a result from the work [31] considering Assumption 3.4 satisfied, and the set $\mathcal{I}^c$ is uniformly chosen from $[N]$, with probability at least $1 - \dfrac{2}{S} - \dfrac{2K}{N^\alpha}$,

$$\mathbb{E}_{\boldsymbol{x}\sim\mathcal{D}}\left[\min_{\boldsymbol{\Pi}}\|\widehat{\boldsymbol{f}}(\boldsymbol{x}) - \boldsymbol{\Pi}\boldsymbol{f}^{(\natural)}(\boldsymbol{x})\|_2^2\right] \leq \frac{c_3(\zeta + \sqrt{TK}\kappa)^2}{T(1 - \sqrt{K}\kappa)^2} + 64T^{-5/8}(2\|\boldsymbol{X}\|_{\mathrm{F}}\mathcal{R}_{\mathcal{F}})^{0.25} + 16\sqrt{\frac{2\log(4S)}{N}}$$

where

$$\zeta^2 = 4\sqrt{2}\beta K\mathfrak{R}(\mathcal{Z}\circ\mathcal{S}) + 8\log(\beta)\sqrt{\frac{2\log(4S)}{S}} = \mathcal{O}\left(\frac{\beta\mathfrak{R}_{\mathcal{S}}\log S}{\sqrt{S}} + \log(\beta)\frac{\sqrt{\log(1/\delta)}}{\sqrt{S}}\right),$$

$$\varphi^2 = MT^\alpha\left(4\sqrt{2}\beta K\mathfrak{R}(\mathcal{Z}\circ\mathcal{S}) + 8\log(\beta)\sqrt{\frac{2\log(4S)}{S}} + 4/S\right)$$

$$\kappa = \varphi + \xi_1 + \xi_2 + \sqrt{K}\xi_1\xi_2,$$

$$\boldsymbol{X} = [\boldsymbol{x}_{n_1},\ldots,\boldsymbol{x}_{n_S}], \quad \text{where } (m_s, n_s) \in \mathcal{S},$$

$$\mathfrak{R}_{\mathcal{S}}^2 = K^2\log(K) + \frac{K\|\boldsymbol{X}\|_{\mathrm{F}}^2\mathcal{R}_{\mathcal{F}}}{S} + CMK\log\sqrt{NM},$$

$$T = N - |\mathcal{I}|.$$

# E    Proof of Theorem 3.6

The proof of Theorem 3.6 follows similar steps in finite case presented in Theorem 3.5, except that we have the exact recovery of $\widehat{\boldsymbol{G}}$ when $S \to \infty$. We repeat the derivation here for completeness.

We first show that the criterion correctly distinguishes the instance-dependent and instance-independent labels.

Let $(\widehat{\boldsymbol{A}}_m$'s, $\widehat{\boldsymbol{e}}_n^{(m)}$'s and $\widehat{\boldsymbol{f}})$ be an optimal solutions as state in Theorem E. From Lemma (3.1), the following holds as the number of observations grows $S$ infinitely large:

$$\widehat{\boldsymbol{g}}_n^{(m)} = \widehat{\boldsymbol{A}}_m\widehat{\boldsymbol{f}}(\boldsymbol{x}_n) + \widehat{\boldsymbol{e}}_n^{(m)} = \boldsymbol{g}_n{}^{\natural}(m) = \boldsymbol{A}_m^{\natural}\boldsymbol{f}^{\natural}(\boldsymbol{x}_n) + \boldsymbol{e}_n^{\natural(m)}, \forall n, m. \qquad (32)$$

Consider the following relation:

$$\boldsymbol{G}^{\natural} = \begin{bmatrix} \boldsymbol{g}_1^{\natural(1)} & \cdots & \boldsymbol{g}_N^{\natural(1)} \\ \vdots & \ddots & \vdots \\ \boldsymbol{g}_1^{\natural(M)} & \cdots & \boldsymbol{g}_N^{\natural(M)} \end{bmatrix} = \begin{bmatrix} \boldsymbol{A}_1^{\natural}\boldsymbol{f}^{\natural}(\boldsymbol{x}_1) + \boldsymbol{e}_1^{\natural(1)} & \cdots & \boldsymbol{A}_1^{\natural}\boldsymbol{f}^{\natural}(\boldsymbol{x}_N) + \boldsymbol{e}_N^{\natural(1)} \\ \vdots & \ddots & \vdots \\ \boldsymbol{A}_M^{\natural}\boldsymbol{f}^{\natural}(\boldsymbol{x}_1) + \boldsymbol{e}_1^{\natural(M)} & \cdots & \boldsymbol{A}_M^{\natural}\boldsymbol{f}^{\natural}(\boldsymbol{x}_N) + \boldsymbol{e}_N^{\natural(M)} \end{bmatrix}. \qquad (33)$$

Here, each column of the matrix $\boldsymbol{G}^\natural$ is given by

$$[\boldsymbol{G}^\natural]_n = \underbrace{\begin{bmatrix} \boldsymbol{A}_1^\natural \\ \vdots \\ \boldsymbol{A}_M^\natural \end{bmatrix}}_{\boldsymbol{W}^\natural \in \mathbb{R}^{MK \times K}} \boldsymbol{f}^\natural(\boldsymbol{x}_n) + \underbrace{\begin{bmatrix} \boldsymbol{e}_n^{\natural(1)} \\ \vdots \\ \boldsymbol{e}_n^{\natural(M)} \end{bmatrix}}_{\boldsymbol{e}_n^\natural \in \mathbb{R}^{MK}}$$

$$\implies [\boldsymbol{G}^\natural]_n = \boldsymbol{W}^\natural \boldsymbol{f}^\natural(\boldsymbol{x}_n) + \boldsymbol{e}_n^\natural, \forall n. \tag{34}$$

Similarly, we can also define

$$\widehat{\boldsymbol{W}} = \begin{bmatrix} \widehat{\boldsymbol{A}}_1 \\ \vdots \\ \widehat{\boldsymbol{A}}_M \end{bmatrix}, \ \widehat{\boldsymbol{e}}_n = \begin{bmatrix} \widehat{\boldsymbol{e}}_n^{(1)} \\ \vdots \\ \widehat{\boldsymbol{e}}_n^{(M)} \end{bmatrix}.$$

From (32), we get that

$$[\boldsymbol{G}^\natural]_n = \widehat{\boldsymbol{W}} \widehat{\boldsymbol{f}}(\boldsymbol{x}_n) + \widehat{\boldsymbol{e}}_n, \forall n. \tag{35}$$

We will use contradiction-based argument towards this. Let us start by considering the case where there exists $i^\star \in \mathcal{I}^c$ as well as $i^\star \in \widehat{\mathcal{I}}^c$ holds. Then, combining (34) and (35) we get

$$\widehat{\boldsymbol{W}} \widehat{\boldsymbol{f}}(\boldsymbol{x}_{i^\star}) = \boldsymbol{W}^\natural \boldsymbol{f}^\natural(\boldsymbol{x}_{i^\star}), \forall i^\star \in \mathcal{I}^c \cap \widehat{\mathcal{I}}^c. \tag{36}$$

We can rewrite the relation (36) as

$$\widehat{\boldsymbol{W}} \widehat{\boldsymbol{F}}(:, \mathcal{I}^c \cap \widehat{\mathcal{I}}^c) = \boldsymbol{W}^\natural \boldsymbol{F}^\natural(:, \mathcal{I}^c \cap \widehat{\mathcal{I}}^c).$$

Since any submatrix formed by $N - 2|\mathcal{I}^c|$ columns of $\boldsymbol{F}^\natural = [\boldsymbol{f}(\boldsymbol{x}_1), \dots, \boldsymbol{f}(\boldsymbol{x}_N)]$ has rank $K$ and that $\mathrm{rank}(\boldsymbol{W}^\natural) = K$, we get that there exists a nonsingular matrix $\boldsymbol{Q} \in \mathbb{R}^{K \times K}$ such that

$$\widehat{\boldsymbol{W}} = \boldsymbol{W}^\natural \boldsymbol{Q}. \tag{37}$$

Next, we consider the case where there exists $i^\star \in \mathcal{I}$ and $i^\star \in \widehat{\mathcal{I}}^c$. Then, again from (34), (35) we get that

$$\widehat{\boldsymbol{W}} \widehat{\boldsymbol{f}}(\boldsymbol{x}_{i^\star}) = \boldsymbol{W}^\natural \boldsymbol{f}^\natural(\boldsymbol{x}_{i^\star}) + \boldsymbol{e}_{i^\star}^\natural \tag{38}$$

However, by applying the relation (37) in (38), we get

$$\boldsymbol{W}^\natural \boldsymbol{Q} \widehat{\boldsymbol{f}}(\boldsymbol{x}_{i^\star}) = \boldsymbol{W}^\natural \boldsymbol{f}^\natural(\boldsymbol{x}_{i^\star}) + \boldsymbol{e}_{i^\star}^\natural$$

$$\implies \boldsymbol{W}^\natural \boldsymbol{Q} \widehat{\boldsymbol{f}}(\boldsymbol{x}_{i^\star}) - \boldsymbol{W}^\natural \boldsymbol{f}^\natural(\boldsymbol{x}_{i^\star}) = \boldsymbol{e}_{i^\star}^\natural$$

$$\implies \boldsymbol{W}^\natural \boldsymbol{q} = \boldsymbol{e}_{i^\star}^\natural,$$

where $\boldsymbol{q} = \boldsymbol{Q} \widehat{\boldsymbol{f}}(\boldsymbol{x}_{i^\star}) - \boldsymbol{f}^\natural(\boldsymbol{x}_{i^\star})$ is a nonzero vector since it is assumed that $\boldsymbol{e}_{i^\star}^\natural \neq \boldsymbol{0}$. The above relation implies that $\boldsymbol{e}_{i^\star}^\natural \in \mathrm{range}(\boldsymbol{W}^\natural)$ must hold which is a contradiction to the assumption that $\boldsymbol{e}_{i^\star}^\natural \notin \mathrm{range}(\boldsymbol{W}^\natural)$ when $i^\star \in \mathcal{I}$. Hence, the relation (38) is feasible only if $\boldsymbol{e}_{i^\star}^\natural = \boldsymbol{0}$. It means that if there exists $i^\star \in \widehat{\mathcal{I}}^c$, we should have $i^\star \in \mathcal{I}^c$, leading to the conclusion that $\widehat{\mathcal{I}}^c \subseteq \mathcal{I}^c$.

On the other hand, if $\widehat{\mathcal{I}}^c \subset \mathcal{I}^c$ holds, then

$$\sum_{n=1}^N \mathbb{1} \left\{ \sum_{m=1}^M \|\widehat{\boldsymbol{e}}_n^{(m)}\|_2 > 0 \right\} > \sum_{n=1}^N \mathbb{1} \left\{ \sum_{m=1}^M \|\boldsymbol{e}_n^{\natural(m)}\|_2 > 0 \right\}.$$

Since $\widehat{\boldsymbol{e}}_n^{(m)}$'s are the optimal solution of Problem (9) with minimal $\sum_{n=1}^N \mathbb{1} \left\{ \sum_{m=1}^M \|\widehat{\boldsymbol{e}}_n^{(m)}\|_2 > 0 \right\}$, the following should hold

$$\sum_{n=1}^N \mathbb{1} \left\{ \sum_{m=1}^M \|\widehat{\boldsymbol{e}}_n^{(m)}\|_2 > 0 \right\} \leq \sum_{n=1}^N \mathbb{1} \left\{ \sum_{m=1}^M \|\boldsymbol{e}_n^{\natural(m)}\|_2 > 0 \right\},$$

which is a contradiction. This implies that $\mathcal{I} = \widehat{\mathcal{I}}$ must hold.

Since $\mathcal{I} = \widehat{\mathcal{I}}$ holds, we have the following relation from (32):

$$G^\natural(:,\mathcal{I}^c) = \widehat{G}(:,\mathcal{I}^c), \tag{39}$$

where we utilized the following notations:

$$\widehat{G}(:,\mathcal{I}^c) = \widehat{W}\widehat{F}(:,\mathcal{I}^c),$$
$$G^\natural(:,\mathcal{I}^c) = W^\natural F^\natural(:,\mathcal{I}^c),$$
$$\widehat{g}_i = [g_i^{(1)\top},\ldots,g_i^{(M)\top}]^\top, i \in \mathcal{I}^c,$$
$$W^\natural = [A_1^{\top\natural},\ldots,A_M^{\top\natural}]^\top$$
$$\widehat{W} = [\widehat{A}_1^\top,\ldots,\widehat{A}_M^\top]^\top.$$

Next, we connect the $\det(W^\top W)$ term to our optimal solutions. This part is similar to the classical proofs in [84, 85], with proper modifications. Consider the following result extracted from the proof of [84, Theorem 1]:

**Lemma E.1.** *Suppose a matrix $Y \in \mathbb{R}^{K \times J}$ satisfies $Y \geq 0$, $\mathbf{1}^\top Y = \mathbf{1}^\top$, rank$(Y) = K$, and SSC. Then, for any $\widehat{Y} = QY$ satisfying $\widehat{Y} \geq 0$, $\mathbf{1}^\top \widehat{Y} = \mathbf{1}^\top$, the following holds:*

$$|\det(Q)| \leq 1,$$

*The equality holds only if $Q$ is a permutation matrix.*

Since $\widehat{W}$ is the optimal solution of the criterion given by Theorem 3.6, the following holds:

$$\det(\widehat{W}^\top \widehat{W}) \leq \det(W^{\natural\top} W^\natural). \tag{40}$$

We also have $W^\natural$ and $F^\natural(:,\mathcal{I}^c)$ are of rank $K$. Hence, from (39), we get that $\widehat{W}$ and $\widehat{F}(:,\mathcal{I}^c)$ satisfies $\widehat{W} = W^\natural Q^{-1}$, $\widehat{F}(:,\mathcal{I}^c) = QF^\natural(:,\mathcal{I}^c)$, for a certain invertible matrix $Q$. Let us assume that $Q$ is not a permutation matrix, Then, we have

$$\det(W^{\natural\top} W^\natural) = \det(Q^\top \widehat{W}^\top \widehat{W} Q)$$
$$= |\det(Q)|^2 \det(\widehat{W}^\top \widehat{W})$$
$$< \det(\widehat{W}^\top \widehat{W}),$$

where the last inequality is from Lemma E.1 using the SSC condition on $F^\natural(:,\mathcal{I}^c)$. Note that the result is a contradiction from (40). Hence, $Q$ must be a permutation matrix. This implies that the optimal solution $\widehat{W}$ and $\widehat{F}(:,\mathcal{I}^c)$ satisfies the following:

$$\widehat{W} = W^\natural \Pi, \tag{41a}$$
$$\widehat{F}(:,\mathcal{I}^c) = \Pi^\top F^\natural(:,\mathcal{I}^c). \tag{41b}$$

## F  Proof of Fact 2.1

For rank$(A^\natural) = K$, when $N \to \infty$, the objective in (6) seeks the optimal solutions $A^\star$, $f^\star$ and $e_n^\star$ such that

$$A^\natural f^\natural(x) + e_n^\natural = A^\star f^\star(x_n) + e_n^\star, \quad n \in [N]. \tag{42}$$

This can be seen from Lemma 3.1 with $M = 1$. One solution that satisfies the equality (42) and the sparsity constraint in (6) is $A^\star = I$, $f^\star(x) = A^\natural f^\natural(x_n) + e_n^\natural$, and $e_n^\star = 0$, which can always hold when $f$ is a universal function approximator.

Table 4: Noise levels and methods used to train machine annotators.

| Case | Noise Level | Training Method/Architecture |
|------|-------------|------------------------------|
| CIFAR-10 | | |
| High Noise | 0.26, 0.69, 0.57 | Logistic Regression, $k$-NN, ResNet34 |
| Medium Noise | 0.26, 0.46, 0.57 | $k$-NN, ResNet34, ResNet34 |
| Low Noise | 0.26, 0.29, 0.57 | $k$-NN, ResNet34, ResNet34 |
| STL-10 | | |
| High Noise | 0.15, 0.25, 0.69 | $k$-NN, ResNet9, ResNet9 |
| Medium Noise | 0.15, 0.36, 0.21 | ResNet9, ResNet9, ResNet9 |
| Low Noise | 0.15, 0.02, 0.21 | ResNet9, ResNet9, ResNet9 |

Table 5: Average classification accuracy on CIFAR-10 using synthetic annotators over 3 random trials.

| Method | Noise rate $\tau = 0.2$ | | | Noise rate $\tau = 0.4$ | | |
|--------|------------|------------|------------|------------|------------|------------|
| | $\eta = 0.1$ | $\eta = 0.3$ | $\eta = 0.5$ | $\eta = 0.1$ | $\eta = 0.3$ | $\eta = 0.5$ |
| CrowdLayer | $91.01 \pm 0.28$ | $88.49 \pm 0.39$ | $84.59 \pm 0.47$ | $89.00 \pm 0.31$ | $86.47 \pm 0.33$ | $82.39 \pm 0.09$ |
| TraceReg | $91.76 \pm 0.38$ | $89.55 \pm 0.20$ | $83.60 \pm 3.14$ | $89.95 \pm 0.19$ | $87.28 \pm 0.08$ | $76.73 \pm 1.24$ |
| MaxMIG | $88.85 \pm 0.29$ | $84.99 \pm 0.41$ | $81.96 \pm 0.16$ | $86.27 \pm 0.59$ | $82.69 \pm 0.57$ | $78.60 \pm 0.33$ |
| GeoCrowdNet(F) | $91.36 \pm 0.26$ | $89.13 \pm 0.29$ | $85.28 \pm 0.34$ | $89.21 \pm 0.43$ | $86.73 \pm 0.36$ | $83.03 \pm 0.49$ |
| GeoCrowdNet(W) | $91.21 \pm 0.28$ | $88.69 \pm 0.29$ | $84.68 \pm 0.44$ | $89.16 \pm 0.14$ | $86.17 \pm 0.39$ | $81.49 \pm 0.31$ |
| MEIDTM | $91.27 \pm 0.09$ | $84.50 \pm 4.24$ | $52.77 \pm 4.05$ | $87.74 \pm 0.09$ | $61.33 \pm 3.92$ | $49.88 \pm 1.11$ |
| PTD | $79.59 \pm 0.46$ | $69.02 \pm 1.61$ | $52.56 \pm 1.05$ | $70.33 \pm 0.63$ | $50.94 \pm 5.48$ | $37.83 \pm 5.87$ |
| BLTM | $73.82 \pm 1.77$ | $66.27 \pm 1.60$ | $53.72 \pm 1.84$ | $66.34 \pm 1.84$ | $55.83 \pm 0.67$ | $46.20 \pm 0.97$ |
| VolMinNet | $89.95 \pm 0.23$ | $87.34 \pm 0.26$ | $82.60 \pm 0.50$ | $87.32 \pm 0.38$ | $83.62 \pm 0.14$ | $76.50 \pm 1.13$ |
| Reweight | $87.39 \pm 0.82$ | $82.35 \pm 0.79$ | $81.82 \pm 0.10$ | $88.34 \pm 0.44$ | $86.16 \pm 0.55$ | $81.67 \pm 0.61$ |
| GCE | $88.75 \pm 0.31$ | $86.98 \pm 0.21$ | $83.72 +/ 0.20$ | $86.70 \pm 0.15$ | $84.58 \pm 0.26$ | $80.29 \pm 0.56$ |
| COINNet (Ours) | $\mathbf{92.23 \pm 0.36}$ | $\mathbf{90.06 \pm 0.06}$ | $\mathbf{86.47 \pm 0.04}$ | $\mathbf{91.97 \pm 0.02}$ | $\mathbf{87.90 \pm 0.03}$ | $\mathbf{83.45 \pm 0.04}$ |

# G    More Details on Experiment Settings and Datasets

## G.1    Implementation

We train with batch size of 512, number of epochs 200, `Adam` optimizer with learning rate of $0.01$ and learning rate scheduler `OneCycleLR` [86]. We initialize $\boldsymbol{A}_m$ using an identity matrix going through softmax and $\boldsymbol{e}_m = \boldsymbol{0}$ for all $m \in [M]$. We also apply standard data augmentation including random cropping, random flipping, when training on CIFAR-10, CIFAR-10N, and STL-10. The experiment results are averaged over 3-5 random trials. All runs have been conducted using either Nvidia A40 or Nvidia DGX H100 GPU. Each run of `COINNet` on CIFAR-10 takes about 60 minutes using Nvidia A40, and 30 minutes using Nvidia DGX H100 GPU.

## G.2    Machine Annotations

For machine annotator experiments, we consider the CIFAR-10 [62] and the STL-10 datasets [63]. The CIFAR-10 dataset consists of $60,000$ labeled color images of animals, vehicles and so on, each having a size of $32 \times 32$ and belonging $K = 10$ different classes. The images are split into training and testing sets with size 50,000 and 10,000, respectively. The STL-10 dataset consists of 13,000 labeled images from 10 different classes, similar to CIFAR-10. Of these, 5,000 images are designated for training and 8,000 for testing.

Table 4 shows the details regarding the individual label noise rates and training methods in our experiments with machine annotators.

## G.3    Real Annotations

For real annotation experiments, we use 2 popular public datasets: CIFAR-10N [66] and LabelMe [67, 68], and construct the ImageNet-15N dataset with a higher number of annotators. All three datasets are labeled by anonymous annotators from the AMT platform. The CIFAR-10N is a "noisy" version of the popular CIFAR-10 dataset, containing 50,000 images for training, and a separate

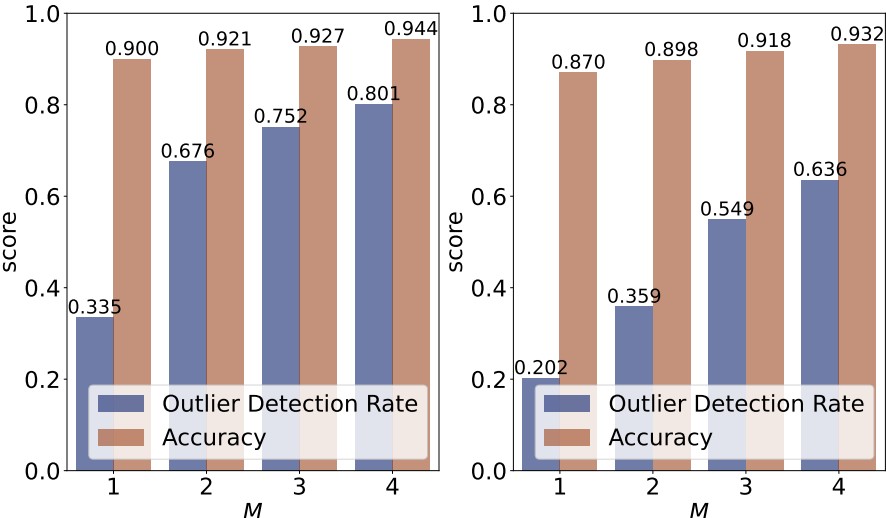

Figure 4: Performance of `COINNet` on CIFAR-10 with synthetic labels against different number of annotators; left: $\tau = 0.2, \eta = 0.1$, right: $\tau = 0.4, \eta = 0.1$.

10,000 images for testing. We randomly split the training set into 47,500 and 2,500 to use as train and validation set for all methods. Every image in CIFAR-10N has 3 labels.

The LabelMe dataset comprises 2,688 images from eight distinct classes: highway, inside city, tall building, street, forest, coast, mountain, and open country. Out of the available images, 1,000 have been annotated by 59 AMT workers. In total, approximately 2,547 image annotations were collected, with labeling accuracy ranging from 0% to 100% and an average accuracy of 69.20%. To enhance the training dataset, standard augmentation techniques such as rescaling, cropping, and horizontal flips were applied, resulting in a training set of 10,000 images annotated by the same 59 workers—see more details in [67]. The validation set comprises 500 images, while the remaining 1,188 images are reserved for testing.

We select 15 classes from the ImageNet dataset with the intention of including easy-to-confuse classes. These classes are: dog, leopard, sports_car, tiger_cat, airship, aircraft_carrier, trailer_truck, orange, penguin, lemon, soccer_ball, airliner, freight_car, container_ship, passenger_car. We collect annotations for $N = 2,514$ images from $M = 100$ anonymous real annotators. These serve as our training set. The average error rate of the annotators is 42.68%. The validation and testing sets have 1,462 and 13,157 images, respectively. We release the noisy annotations at `https://github.com/ductri/COINNet`.

## H  More Experiments

### H.1  Experiments with Synthetic Annotators

**Noisy Label Generation.** Here, we consider labels corrupted by synthetically generated annotator confusions. To generate the confusion matrices, we utilize the following strategy. We control the instance-independent label noise rate using a parameter $\tau \in (0,1)$. By employing this parameter, we construct the ground-truth confusion matrix $\boldsymbol{A}^\natural$, where the diagonal entries of $\boldsymbol{A}^\natural$ are set as $[\boldsymbol{A}^\natural]_{k,k} = 1 - \tau, \forall k$, and the off-diagonal entries are set as $[\boldsymbol{A}^\natural]_{k,j} = \frac{\tau}{K-1}, \forall k \neq j$. First, we randomly select $1 - \eta$ fraction of the data samples and generate their labels using the instance-independent confusion matrix $\boldsymbol{A}^\natural$. For the remaining $\eta$ percent of the samples, we adopt an instance-dependent noise generation process, by following the strategy outlined in [27]. Specifically, for such "outliers", the label confusions vary depending on the features of such instances.

Table 6: Average classification accuracy on CIFAR-10 and STL-10 datasets using machine annotators; results are averaged over 3 random trials.

| Method | CIFAR-10 | | | STL-10 | | |
|---|---|---|---|---|---|---|
| | High Noise | Medium Noise | Low Noise | High Noise | Medium Noise | Low Noise |
| COINNet, $\mu_1 = 0.01, \mu_2 = 0.01$ | **71.22 ± 0.72** | **73.31 ± 0.09** | **84.14 ± 0.38** | **70.12 ± 0.48** | **73.11 ± 0.37** | **76.39 ± 0.58** |
| COINNet, $\mu_1 = 0.01, \mu_2 = 0.00$ | 71.04 ± 0.33 | 73.11 ± 0.31 | 83.61 ± 0.68 | 58.90 ± 3.44 | 64.47 ± 3.75 | 72.79 ±2.88 |
| COINNet, $\mu_1 = 0.00, \mu_2 = 0.01$ | 71.17 ± 0.46 | 73.18 ± 0.91 | 83.51 ± 0.47 | 66.77 ± 4.81 | 72.40 ± 0.57 | 76.24 ± 0.52 |

Table 7: Average classification accuracy on CIFAR-10 using synthetic annotators; results are averaged over 3 random trials.

| Method | Noise rate $\tau = 0.2$ | | | Noise rate $\tau = 0.4$ | | |
|---|---|---|---|---|---|---|
| | $\eta = 0.1$ | $\eta = 0.3$ | $\eta = 0.5$ | $\eta = 0.1$ | $\eta = 0.3$ | $\eta = 0.5$ |
| COINNet, $\mu_1 = 0.01, \mu_2 = 0.01$ | **92.23 ± 0.36** | **90.06 ± 0.06** | **86.47 ± 0.04** | **91.97 ± 0.02** | 87.90 ± 0.03 | **83.45 ± 0.04** |
| COINNet, $\mu_1 = 0.01, \mu_2 = 0.00$ | **92.23 ± 0.93** | 89.64 ± 0.16 | 86.40 ± 0.18 | 91.16 ± 0.07 | **87.98 ± 0.15** | 83.15 ± 0.16 |
| COINNet, $\mu_1 = 0.00, \mu_2 = 0.01$ | 91.87 ± 0.04 | 89.65 ± 0.13 | 85.44 ± 0.31 | 90.21 ± 0.22 | 87.01 ± 0.48 | 66.50 ± 1.44 |

Table 8: Average classification accuracy over 3 random trials on real datasets.

| Method | CIFAR-10N |
|---|---|
| COINNet, $\mu_1 = 0.01, \mu_2 = 0.01$ | 92.09 ± 0.47 |
| COINNet, $\mu_1 = 0.01, \mu_2 = 0.00$ | **92.26 ± 0.24** |
| COINNet, $\mu_1 = 0.00, \mu_2 = 0.01$ | 91.51 ± 0.60 |
| **Method** | LabelMe |
| COINNet, $\mu_1 = 0.10, \mu_2 = 0.10$ | **87.60 ± 0.54** |
| COINNet, $\mu_1 = 0.10, \mu_2 = 0.00$ | 83.64 ± 1.29 |
| COINNet, $\mu_1 = 0.00, \mu_2 = 0.10$ | 84.85 ± 0.22 |

We generate synthetic noisy labels for CIFAR-10 dataset with $M = 3$. CIFAR-10 contains 50,000 images of size $32 \times 32$ as training set, and 10,000 images as test set. We keep the test set untouched while randomly splitting the train set into 2 parts (47,500; 2,500) to serve as training and validation set. We use the same neural network architecture and the optimization settings as used in machine annotator case. The results are presented in Table 5. One can observe that our approach consistently perform well in all scenarios.

To demonstrate the effectiveness of the advocated crowdsourcing approach in outlier detection, we vary the number of annotators $M$ and present the average outlier detection rate and the testing accuracy, where

$$\text{Outlier Detection Rate} = \frac{|\mathcal{I} \cap \{i \mid \|\widehat{e}_i\|_2 \in \text{top } |\mathcal{I}| \text{ values among all } \|\widehat{e}_n\|_2, n \in [N]\}|}{|\mathcal{I}|}.$$

We observe that increasing the number of annotators $M$ shows improvement over both outlier detection and the final accuracy score as shown in Fig. 4.

## H.2 Ablation Study

**Effect of regularization terms.** Tables 6, 7 and 8 show the effectiveness of having both the proposed regularization terms (i.e., sparsity and volume).

**Effect of missing annotations.** Table 9 and Table 10 show results under different levels of missing annotations under various parameters settings. Table 11 and Table 12 present results where only a single label is available for each image. In all four cases, the proposed method demonstrates superior robustness against the negative effect of missing annotations relative to MaxMIG and GeoCrowdNet(F), two competitive baselines.

**Effect of initialization.** In Table 13, we test the performance of our approach using different initialization strategies. In particular, we conduct the machine annotator experiment on CIFAR-10 using the following initialization strategies: the confusion matrices are (i) initialized using identity matrices, and (ii) initialized by the GeoCrowdNet(F) method. We also include the performance of GeoCrowdNet(F) for reference. We observe a slight improvement (around 0.1-0.4%) in accuracy when using initialization from GeoCrowdNet(F) with the cost of training GeoCrowdNet(F) for 10 epochs.

Table 9: Average classification accuracy v.s. missing rate on CIFAR-10 using synthetic annotators; $M = 3, \tau = 0.2, \eta = 0.3$; results are averaged over 3 random trials.

| missing rate | 0.1 | 0.2 | 0.3 | 0.4 | 0.5 |
|---|---|---|---|---|---|
| MaxMIG | $85.01 \pm 0.26$ | $84.32 \pm 0.20$ | $83.90 \pm 0.59$ | $83.17 \pm 0.70$ | $81.80 \pm 0.25$ |
| GeoCrowdNet(F) | $83.19 \pm 0.63$ | $83.03 \pm 0.41$ | $81.83 \pm 0.17$ | $81.46 \pm 0.47$ | $81.28 \pm 0.56$ |
| COINNet(Ours) | $\mathbf{89.41 \pm 0.13}$ | $\mathbf{89.22 \pm 0.21}$ | $\mathbf{88.98 \pm 0.55}$ | $\mathbf{88.53 \pm 0.40}$ | $\mathbf{87.54 \pm 0.16}$ |

Table 10: Average classification accuracy v.s. missing rate on CIFAR-10 using synthetic annotators; $M = 3, \tau = 0.2, \eta = 0.5$; results are averaged over 3 random trials.

| missing rate | 0.1 | 0.2 | 0.3 | 0.4 | 0.5 |
|---|---|---|---|---|---|
| MaxMIG | $80.97 \pm 0.77$ | $79.04 \pm 0.90$ | $79.46 \pm 0.62$ | $77.78 \pm 0.72$ | $76.15 \pm 1.81$ |
| GeoCrowdNet(F) | $80.32 \pm 0.55$ | $79.42 \pm 0.27$ | $78.81 \pm 0.27$ | $76.93 \pm 0.39$ | $76.01 \pm 0.42$ |
| COINNet(Ours) | $\mathbf{86.23 \pm 0.14}$ | $\mathbf{84.98 \pm 0.64}$ | $\mathbf{84.24 \pm 0.71}$ | $\mathbf{83.79 \pm 0.40}$ | $\mathbf{81.49 \pm 0.59}$ |

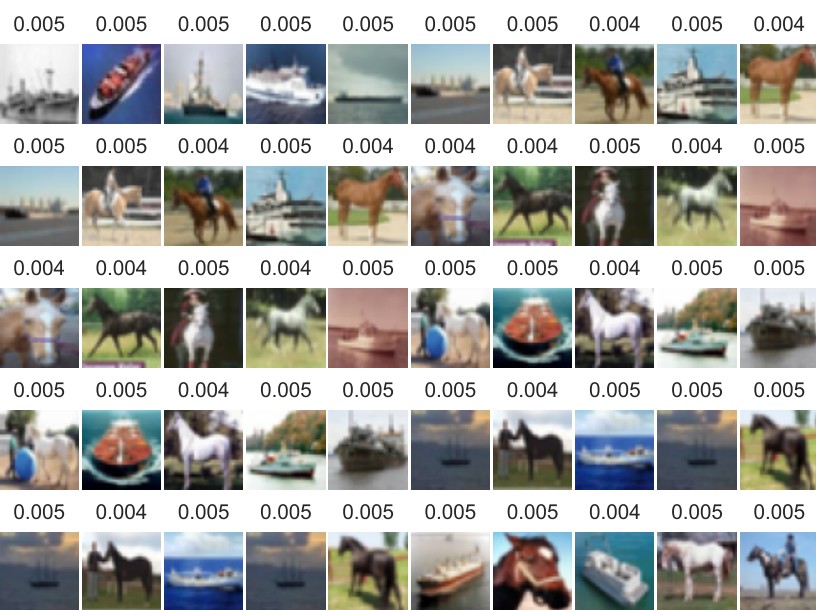

Figure 5: Some examples from the CIFAR-10N dataset learned with low outlier scores $s_n$'s.

### H.3 Outlier Detection in CIFAR-10N Dataset

Following the result in Fig. 2, we show more examples of nominal data items and outlier data items as separated by our approach in Fig. 5 and Fig. 6, respectively. Clearly, more instance-varying confusing characteristics are present in the latter case.

Table 11: Average classification accuracy in the scenario where each image of CIFAR-10 is labeled by only one randomly chosen annotator from $M = 3$ synthetic annotators; results are averaged over 3 random trials.

|  | $\tau = 0.2, \eta = 0.3$ | $\tau = 0.2, \eta = 0.5$ |
|---|---|---|
| MaxMIG | $81.36 \pm 0.81$ | $70.78 \pm 4.66$ |
| GeoCrowdNet(F) | $80.44 \pm 0.18$ | $75.31 \pm 0.33$ |
| COINNet(Ours) | $\mathbf{87.18 \pm 0.29}$ | $\mathbf{80.96 \pm 0.38}$ |

Table 12: Average classification accuracy in the scenario where each image of CIFAR-10 is labeled by only one randomly chosen annotator from $M = 3$ machine annotators; results are averaged over 3 random trials—see the annotation generation settings as in Sec. 5.1.

|  | High Noise | Medium Noise | Low Noise |
|---|---|---|---|
| MaxMIG | $58.58 \pm 0.64$ | $67.64 \pm 0.25$ | $77.01 \pm 0.25$ |
| GeoCrowdNet(F) | $55.25 \pm 1.09$ | $66.91 \pm 0.53$ | $75.48 \pm 0.28$ |
| COINNet(Ours) | $\mathbf{59.76 \pm 0.62}$ | $\mathbf{70.94 \pm 0.32}$ | $\mathbf{80.12 \pm 0.44}$ |

Table 13: Average classification accuracy with different initializations for the confusion matrices. We use machine annotations with the same setting as described in Sec. 5.1 in the manuscript. Results are averaged over 3 random trials.

| Initialization Strategy for $A_m$'s | High Noise | Medium Noise | Low Noise |
|---|---|---|---|
| Identity matrix | $70.37 \pm 0.61$ | $73.33 \pm 0.46$ | $83.25 \pm 0.39$ |
| GeoCrowdNet(F) after training 10 epochs | $\mathbf{71.75 \pm 0.73}$ | $\mathbf{73.48 \pm 0.27}$ | $\mathbf{84.26 \pm 0.13}$ |
| Current setting (close to an identity matrix) | $71.22 \pm 0.72$ | $73.31 \pm 0.09$ | $84.14 \pm 0.38$ |

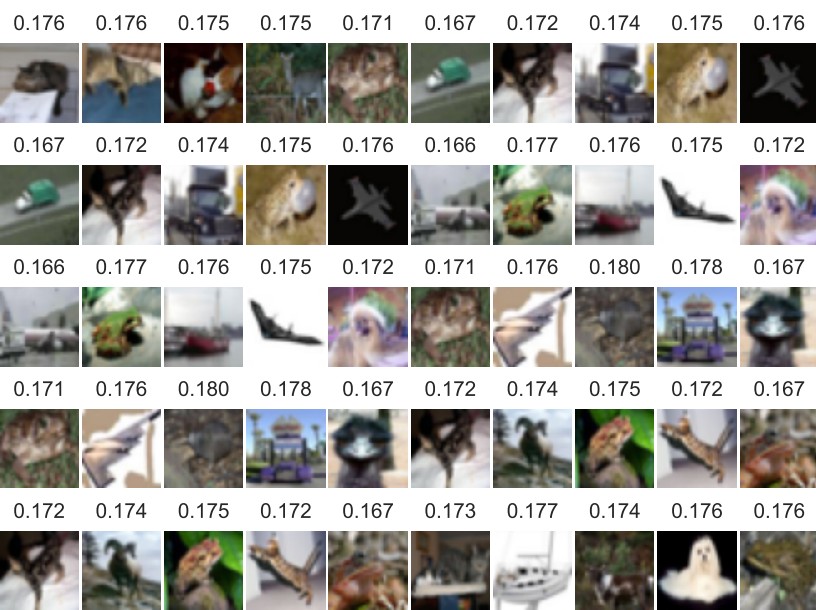

Figure 6: Some examples from the CIFAR-10N dataset learned with high outlier scores $s_n$'s

