# OpenReview forum: "Noisy Label Learning with Instance-Dependent Outliers: Identifiability via Crowd Wisdom"
_NeurIPS.cc/2024/Conference — NeurIPS 2024 spotlight_

### Official Review · Reviewer_BkAV · 2024-07-05

**Soundness:** 3
**Presentation:** 3
**Contribution:** 3
**Rating:** 7
**Confidence:** 3

**Summary:**

The paper addresses the problem of learning with noisy labels by considering a model where instance-dependent confusion matrices occur occasionally across the samples, and the rest of data share a common nominal confusion matrix. The paper claims two main contributions: (1) showing that a single confusion matrix is insufficient to identify outliers and proposing a crowdsourcing strategy with a column sparsity constraint to overcome this, and (2) presenting an end-to-end, one-stage learning loss that is differentiable and easily optimized. These contributions are validated through experiments that demonstrate improved testing accuracy.

**Strengths:**

● The problem setting, considering the instance-dependent noisy samples as outliers, is realistic and really interesting.
● The paper posits that existing methods relying on sparsity priors are insufficient for outlier detection and proposes a novel crowdsourcing strategy as a solution with theoretical grounding and generalization guarantees.

**Weaknesses:**

Lack of large-scale datasets in experiments

**Questions:**

Can the authors discuss any limitations their method might have in scenarios of highly imbalanced noise?

---

> ### Author Rebuttal · Authors · 2024-08-07
>
> __[Regarding Large-Scale Datasets]__ We agree with the reviewer on this point. In the current manuscript, we have tested our methods against baselines on a dataset of size 50,000 samples in both machine annotations and real datasets. We did not use larger data size due to resource limitations and also because our focus has been on the design principle side.
> Nonetheless, to examine the proposal’s scalability, we have conducted an experiment on the full SVHN dataset which contains more than 600,000 images. In this experiment, $K=10$. We used $M=3$ machine annotators as we defined in the manuscript. The error rates of the annotators are 0.30, 0.36, and 0.32. We also included performance of $\texttt{GeoCrowdNet(F)}$ and $\texttt{MaxMIG}$ for comparison. The result is presented in _Table 6 in the attached pdf_. Although we have not been able to sufficiently tune our method’s parameters due to time limitation, the preliminary result already looks promising.
>
> __[Regarding Imbalanced Noise]__ Thank you for this question. Please note that Table 1 in the manuscript addresses the imbalanced noise setting, where annotators have different annotation noise rates. Table 4 of the Appendix G presents the individual noise rates for each scenario. As shown in the table, in the high level noise case, the noise rates for the three annotators are 26%, 69%, 57% for CIFAR-10, and 15%, 25%, 69% for STL-10, respectively. The results in Table 1 show that our method is robust to such highly imbalanced noise settings, outperforming the second-best baseline by approximately 3% on average.

---

### Official Review · Reviewer_FFyM · 2024-07-11

**Soundness:** 3
**Presentation:** 3
**Contribution:** 3
**Rating:** 5
**Confidence:** 5

**Summary:**

The paper investigates the challenge of instance-dependent noisy labels, which are modeled using an instance-dependent confusion matrix reflecting annotator errors. Traditional approaches assume a consistent confusion matrix across instances, which simplifies the problem but is unrealistic. This study models the instance-dependent confusion matrices as outliers. The authors propose using a crowdsourcing strategy with multiple annotators and a specialized loss function to effectively detect outliers and identify the target classifier. The method is supported by extensive theoretical results. Experimental results also confirm the efficacy of the proposed method.

**Strengths:**

1. The work aims at an important and the proposed solution is solid.
2. The presentation is clear and the proposed method is supported by thorough theoretical results.
3. Experiments on real-world label noise such as CIFAR-N validate the effectiveness of the proposed method.

**Weaknesses:**

1. Theorem 3.6 requires $S\rightarrow \infty$. However, we know if we have sufficient annotators, the aggregated noise rate could be 0, making the result trivial. It is interesting to show the result with finite $S$.
2. The organization part of the proposed method could be polished.

**Questions:**

1. Does this method heavily rely on multiple annotators? If there is only one label for each example, is there any approximation method to generate extra labels to simulate multiple annotators?
2. Would existing estimators for noisy labels, such as [R1], provide a good initialization for the variable $\mathbf A_m$ in the proposed loss function?

[R1] Unmasking and improving data credibility: A study with datasets for training harmless language models. ICLR 2024.

---

> ### Author Rebuttal · Authors · 2024-08-07
>
> __[Regarding $S \to \infty$ in Theorem 3.6]__ We agree that an analysis with finite $S$ is more desirable (as we did in Theorem 3.5). Nonetheless, we argue that even with infinite $S$, the result in Theorem 3.6 is meaningful and nontrivial. Let us explain.
> First, the aggregated noise will not disappear when $S \to \infty$. Note that $S$ represents the number of annotations (not the number of annotators). When ($M,N$) are fixed, $S$ can still go to $\infty$ as sampling for annotation is assumed to be with replacement. Recall that for $\textbf{y}_n^{(m)} \sim \textbf{p}_n^{(m)} = \textbf{A}_m \textbf{f}(\textbf{x}_n) + \textbf{E}_n^{(m)}$.
> Even when there is no outliers, if $\textbf{A}_m$ is heavily non-diagonal, $\textbf{p}_n^{(m)}$ is going to be very different from $\textbf{f}(\textbf{x}_n)$, and thus $\textbf{y}_n^{(m)}$ is always going to be wrong no matter how large is $S$.
>
> One possibility of using increased annotators to attain zero aggregated noise is when $\sum_{m=1}^M \textbf{A}_m = \textbf{I}$, but this requires that all the annotator errors are zero mean across $m$---which might be too special to assume. Majority voting also does not help if most $\textbf{A}_m$ are not close to diagonal.
>
> Second, the identifiability analysis is not trivial and the result has good practical implications. Note that $S \to \infty$ makes the equality $\textbf{p}_n^{(m)} = \textbf{A}_m \textbf{f}(\textbf{x}_n) + \textbf{O}_n^{(m)}$ hold for every $m$ and $n$. But this equality per se still does not guarantee the identifiability of any of $\textbf{A}_m$, $ \textbf{f}(\textbf{x}_n)$, or the outlier $\textbf{O}_n$. The point made by Theorem 3.6 is that our formulation with a volume maximization constraint can identify all the three terms. This theorem gives useful insight: when we have sufficiently large $S$, using the outlier sparsity and volume regularization (as shown in line 227-228, page 6) is expected to give good performance. We believe this insight itself can benefit practical implementations.
>
> __[Organization of the Proposed Method Section]__ We will do a thorough proofreading and polish the section.
>
> __[Regarding Multiple Annotators and the One-Label-Per-Item Case]__ This is an interesting question. The method does require the existence of multiple annotators, but it does not require each item to be labeled by all annotators (see the settings of Lemma 3.1 and Theorem 3.5). Therefore, even if each item is annotated by only one annotator, the method still works under reasonable conditions.  To show this, we present some empirical results in _Table 3 and 4 of the attached pdf_. The results suggest that the proposed method still outperforms the competing baselines in this setting.
>
> __[Initialization for A_m]__ Good initialization techniques are always appreciated. We will add discussion in the revised version on initializing $\textbf{A}_m$ using existing methods such as those in [R1]. In _Table 5 of the attached pdf_, we also compared the performance of our approach using different initialization strategies. In particular, we compare the machine annotator experiment on CIFAR-10 using different initialization strategies: (i) $\textbf{A}_m$ initialized using identity matrices and (ii) $\textbf{A}_m$ initialized by the$ \texttt{GeoCrowdNet(F)} $method. We also include the performance of $\texttt{GeoCrowdNet(F)}$ for reference. We observe a slight improvement (around 0.1-0.4%) in accuracy when using initialization from $\texttt{GeoCrowdNet(F)}$ with the cost of training $\texttt{GeoCrowdNet(F)}$ for 10 epochs.

---

### Official Review · Reviewer_TTRF · 2024-07-12

**Soundness:** 3
**Presentation:** 3
**Contribution:** 3
**Rating:** 7
**Confidence:** 4

**Summary:**

The paper extends further estimation of transition matrix in label noise learning. In previous studies, label noise is often assumed to be class-dependent. Hence, one can use the noisy label data and apply loss correction to learn a good classifier. The paper extends from such a modelling approach by considering most of samples having class-dependent label noise and some samples having class- and instance-dependent noise. Those with instance-dependent label noise are then considered as outlier in the newly proposed modelling approach. Despite such a modelling, it still results in a non-identifiable solution. To overcome that, the paper incorporates multiple labels from multiple annotators, so that one can solve for the nominal transition matrix and the classifier of interest. The paper also provides theoretical results to guarantee that the estimation holds under certain confidence level. Empirical results show that models trained on the new loss function performs competitively with prior methods.

**Strengths:**

- The paper thoroughly presents the motivation and the formulation of their modelling approach. The paper also connects to previous studies when formulating the noisy label learning with instance-dependent noise as outliers. In general, the paper is well-written and easy to follow.
- The main contribution of the paper is to extend further the modelling considering only class-dependent transition matrix by adding *perturbation* induced by instance-dependent label noise samples. The paper also analyses the theoretical guarantee when estimating the classifier of interest in terms of PAC-style formulae.

**Weaknesses:**

The major weakness of the paper may lie at the assumption that majority of samples would share the same nominal transition matrix, while a few would have their own transition matices. However, this does not affect too much to the contribution of the paper.

Another weakness is the usage of multiple annotators, while the conventional noisy label learning does not consider any additional labels. Of course, some studies (e.g., [43]) show that using a single noisy label would result in non-identifiability, unless additional constraints are exploited. Nevertheless, multiple annotator setting would be closer to current practice. The downside is that there are not many multi-rater learning datasets for benchmarking.

### Minors
- Line 76: typo "traget" -> target
- The abbreviation "NMF" at line 100 is not defined.
- Line 111: $E(x_{n})$ is a matrix. Thus, please make it clear when specifying $E(x_{n}) > 0$.

**Questions:**

I do enjoy reading the paper. It is coherent and easy to follow. I do not have any questions.

**Limitations:**

As mentioned in the weakness section above and in the conclusion of the paper, the paper only considers a few samples with instance-dependent label noise, while the remaining samples are class-dependent label noise, to reduce the complexity of the modelling and analysis. Such an assumption may not always hold in all settings.

---

> ### Author Rebuttal · Authors · 2024-08-07
>
> Thank you for the insightful comments.
>
>
> __[Regarding the Lack of Public Multi-rater Dataset]__ We agree. It would benefit the community if more public multi-rater datasets are available. We are working on creating such data using Amazon Mechanical Turk (AMT), but the size of data is still limited by resource/cost and it might be beyond the timespan of this submission.
> A relevant remark is that, to conduct experiments under realistic multi-annotator settings, we considered machine annotators in our experiments (see our settings in Sec. 5.1, and Appendix G.2). Note that a machine annotator is a classifier (e.g., SVM,  kernel SVM and KNN based classifiers) that are trained using limited data. Hence, they all make mistakes in “labeling” data. The setting is of interest: (i) integrating the annotations from such erroneous classifiers is called ensemble learning, which is a special case of noisy label learning and crowdsourcing---and is a real application in practice. (ii) The classifiers make mistakes according to the hardness of each item, and thus the setup well represents instance-dependent errors.
> We will release the code for making such machine annotator-based datasets, so that the community could easily create multi-rater scenarios with specified error rates and missing label proportions, etc, to facilitate reproducible research in an economical way.
>
> __[Minor Points]__ Thanks for the attentive reading. We will fix the typos. In particular, $\textbf{E}^{\natural}(\textbf{x}_n)>0$ should be $\textbf{E}^{\natural}(\textbf{x}_n) \neq \textbf{0}$, that is, the matrix is not an all-zero matrix.

---

> > ### Comment · Reviewer_TTRF · 2024-08-14
> > **Comments by Reviewer TTRF**
> >
> > Thank you, the authors, for discussing the concerns I raised in my initial review. Regarding the public datasets, there are a number of available datasets that could be used in noisy label learning and multi-rater learning:
> > - 10 datasets mentioned in the NeurIPS 2022 paper: *Is one annotation enough? - A data-centric image classification benchmark for noisy and ambiguous label estimation*
> > - *dopanim: A Dataset of Doppelganger Animals with Noisy Annotations from Multiple Humans* available on zenodo with DOI 10.5281/zenodo.11479590
> >
> > In addition, I have just found another (preprint) paper investigating the identifiability of the noisy label learning: *Towards the Identifiability in Noisy Label Learning: A Multinomial Mixture Approach*. In that paper, they showed that at least $2C - 1$ noisy labels per sample needed to make the label noise problem identifiable, where $C$ is the number of classes. This is, to me, more practical than [43] because the result of 3 noisy labels per sample in [43] is too optimistic, although in their paper, they also mentioned that it must be larger than 3 (but not explicitly say how many specifically). Hence, I think that it worths to add that into the discussion, especially the authors already showed that the more annotations, the better the estimation.
> >
> > In summary, I believe that this is a good paper in the field of noisy label learning.

---

### Official Review · Reviewer_okUb · 2024-07-15

**Soundness:** 3
**Presentation:** 3
**Contribution:** 3
**Rating:** 6
**Confidence:** 4

**Summary:**

This paper studied the identifiability problem of instance-dependent label noise with multiple annotators. To achieve the identifiability, this work first claimed a fact that only a proportion of all instances may have a labeling difficulty that significantly deviates from the general population. Then, it connected the problem to the uniqueness of non-negative matrix factorization with mild assumptions. Inspired by the identifiability result, this work proposed a end-to-end one-stage method to learn from crowds via identifying instance-dependent label noise. Experiments on multiple datasets with machine annotations and human annotators showed the effectiveness of the proposed method.

**Strengths:**

1. This work provided some interesting identifiability results for learning from crowds, which may inspired further work.
2. The writing is excellent, making it easy to understand.
3. The proposed method is end-to-end and one-stage, which is nice for application.
4. The case study in Figure 2 clearly showed the rationality of the results.

**Weaknesses:**

1. As this work claimed, the number of annotators is important to identify the instance-dependent label noise, is there some experimental results verifying this analysis？

2.  For human annotations,  the annotations are usually sparse. Does the annotation sparsity level influence the identifiability and the performance of the proposed method? I suggest the authors do some experiments as existing works to clarify it [1-3].

3. The meaning of some terms, like "outliers", "the model of interest", "neural systems", is not clear when just reading the abstract. Besides, since the sparsity prior of the outliers is a basis assumption for this work, it should be referred to in the abstract.

[1] Label correction of crowdsourced noisy annotations with an instance-dependent noise transition model. NeurIPS 2023

[2] Coupled confusion correction: learning from crowds with sparse annotations. AAAI 2024

[3] Trustable co-label learning from multiple noisy annotators. TMM 2023

**Questions:**

1. The meaning of some terms, like "outliers", "the model of interest", "neural systems", is not clear when just reading the abstract.
2. It seems that there is a symbol that is not displayed in the restrictive condition of Eq.(6).
3. What does $e^\star_n$ mean in Line 129?
4. What does "over-canceling” mean in Line 211?
5. I suggest placing the learning-from-crowds methods after the learning-from-single-annotator methods in Table 1 and 2.

**Limitations:**

See above.

---

> ### Author Rebuttal · Authors · 2024-08-07
>
> __[Regarding Impact of the Number of Annotators]__ In Figure 3 of the supplementary section H.1, we presented experiments showing the impact of the number of annotators for outlier identification. The results indicate that increasing the number of annotators improves the detection of instance-dependent samples as well as the classification performance. We will add more obvious pointers to this result in the supplementary material.
>
> __[Regarding "Sparse Annotations"]__ Thanks for this suggestion. We hope to remark that our identifiability analysis does cover the “sparse annotation” case—see Eq. 10, Lemma 3.1 and Theorem 3.5 where the bounds depend on $S$, i.e., number of observed labels. Here, $S$ can be much smaller than $NM$, where $N$ is the number of data items and $M$ is the number of annotators. We also thank the reviewer for providing the papers addressing sparse annotations. We will discuss them in “Related Works”. The interest on the sparse annotations also attests to the importance of our theoretical result in Theorem 3.5.
> In our real-data experiments, the LabelMe dataset has many missing labels (please see Table 2 of the manuscript). To be specific, only 2547 annotations are obtained from 59 workers for 1000 images, which implies that 95% of the annotations are missing ($S/(NM) \approx 0.05$). Our approach scores the best classification performance in this case as shown in Table 2 of the manuscript.
> To further analyze the effect of missing annotations, we followed the reviewer’s suggestion to conduct the following synthetic data experiments for various levels of sparsity in the CIFAR-10 dataset—see _Table 1, 2 in the attached pdf_. The results show that the proposed method is more robust to the negative effect brought upon by the missing annotations, relative to the best-performing baselines, namely, $\texttt{MaxMIG}$ and $\texttt{GeoCrowdNet}$.
>
> __[Regarding Abstract]__ Thank you for your suggestions. We agree that the terminologies can be simplified/unified to improve clarity. Mentioning the sparsity prior also makes sense to us. We will revise our abstract accordingly.
>
> __Questions:__
>
> __[Q1]__ Agreed. We will revise accordingly.
>
> __[Q2]__ The symbol $\mathbb{I}[E]$ in the constraint denotes the indicator function, whose value is 1 if the event $E$ happens and 0 otherwise. We defined this notation in Table 3 in Appendix A. To avoid confusion, we will make this clearer in the main text.
>
> __[Q3]__ $(\textbf{A}_n^\star, \textbf{e}_n^\star, \textbf{f}^\star)$ denotes an optimal solution of Problem (6). We will make it clearer.
>
> __[Q4]__  Ideally, we hope to just cancel/exclude the exact set of outliers from the process of learning $\textbf{f}$. This would require us to set $C=\| \mathcal{I}\|$ in Problem (6). Nonetheless, the exact number $\|\mathcal{I}\|$ is hard to estimate. Our analysis in Theorem 3.6 shows that for even when $C$ is over-estimated ( i.e., $C > \|\mathcal{I}\|$), our method still can cancel out all the outliers. As a price to pay for overestimating $C$,  there will be $C- \|\mathcal{I}\|$ nominal data samples also identified as outliers and excluded for learning $\textbf{f}$. This is what we meant by “over-canceling”.
> Thanks for pointing out this ambiguity. We will rephrase this sentence to make this discussion clearer.
>
> __[Q5]__ Thank you for your suggestion. We will do the rearrangement in the revised version.

---

### Author Rebuttal · Authors · 2024-08-07

We would like to thank all reviewers for their attentive reading and valuable comments/suggestions. We have replied to each reviewer in their corresponding sections. Here we present a summary of major comments and our responses accordingly.

__[Reviewer okUb]__
Reviewer okUb suggested investigating the impact of annotation sparsity and the number of annotators. We clarified that the effects of both annotation sparsity and number of annotators are covered by our analysis and some existing experiments. To further observe these aspects, we have additionally run more experiments under various sparsity levels and presented the results in _Table 1 and 2 in the attached pdf_. We also clarified a number of notation and terminology questions.

__[Reviewer TTRF]__
Reviewer TTRF made a comment regarding the lack of public multi-rater datasets in this field.  We followed up with some thoughts and offered our perspectives. We highlighted the benefits of including machine annotators in our experiments and its usefulness in facilitating reproducible, realistic experiments in an economical way.


__[Reviewer FFyM]__
Firstly, the reviewer FFyM raised concerns about the significance of the infinite sample analysis in Theorem 3.6. In our response, we clarified the merit of Theorem 3.6 under the infinite case, when the number of annotations goes to infinity. We pointed out that the aggregated noise would not be canceled out by simply letting $S \to \infty$ and explained the practical implications and significance of the identifiability analysis of Theorem 3.6.

Secondly, reviewer FFyM asked about the importance of multiple annotators and suggested considering a one-label-per-sample case. We clarified that our method is able to work with such single-label data and provided additional results in _Table 3 and 4 in the attached pdf_.

Lastly, we followed reviewer FFyM’s suggestion on trying different initializations, and provided additional empirical study (in _Table 5 of the attached pdf_).

__[Reviewer BkAV]__
Reviewer BkAV made a comment about experiments using large-scale datasets. We replied and offered our perspectives on these points. To address this comment, we conducted an experiment on the full SVHN dataset which contains more than 600k images. The result is presented in _Table 6 in the attached pdf_.

In addition, reviewer BkAV asked about the performance of the proposed method under imbalanced annotation noise. We pointed out that we had already considered such a setting in our experiments and provided a pointer to the pertinent results in the appendix.

---

### Decision · Program_Chairs · 2024-09-25

**Decision:**

Accept (spotlight)

**Comment:**

There is a clear consensus in the PC that this is a strong paper that should be accepted. Congratulations to the authors! I am requesting the author to carefully incorporate the suggestion provided by the review in the camera-ready version.